# SteinDiff: Resolving the Contractivity Trap via Reference-free Stein Regularization

## Abstract

A fundamental tension arises when accelerating diffusion-based generative models via their deterministic probability flow ordinary differential equation (PF-ODE) paths, which we formally identify as the "contractivity trap": efficient inference requires large step sizes, but stable convergence demands strong contractivity that limits model expressiveness. This results in error accumulation in inference as contractivity weakens. In this work, we propose a principled inference approach, called *SteinDiff*, that relaxes the contractivity constraints through reference-free Stein regularization. Specifically, drawing on Krasnosel'skiǐ-Mann theory, we reformulate the discretized ODE update operator to interpolate between predictions and current states. Importantly, we contribute efficient closed-form regularization estimators via Stein's identity, which is grounded in the continuous SDE theory of diffusion models. Our step-wise analytical approach eliminates the need for ground truth data to adapt to the local geometry of the data distribution while preserving the expressiveness of the vanilla model. Theoretically, our approach not only relaxes the strict contractivity requirements for robust convergence but also reveals a principle behind the stability of state-of-the-art (SOTA) pre-conditioned parameterizations. Practically, we offer a reference-free solution that reduces the risk of mode collapse in large-step inference. Extensive experiments validate our theoretical framework and demonstrate significant gains in generative inference.

## 1 Introduction

Diffusion models (DMs) have revolutionized generative modeling, achieving unprecedented quality in image synthesis, text-to-image generation, and other domains (Sohl-Dickstein et al., 2015; Ho et al., 2020; Song et al., 2021b; Dhariwal & Nichol, 2021; Rombach et al., 2022; Xing et al., 2024). Unlike single-pass generators such as GANs (Goodfellow et al., 2014) and VAEs (Kingma, 2013), DMs generate samples through an iterative denoising process that progressively transforms noise into structured data. This iterative paradigm provides advantages: training stability, high sample quality, and robust mode coverage (Song & Ermon, 2020; Kingma et al., 2021; Karras et al., 2022).

Despite these advantages, the inference process suffers from a critical limitation. High-quality samples typically require hundreds of function evaluations (NFE) (Sohl-Dickstein et al., 2015; Ho et al., 2020). Although ODE-based implicit models enable acceleration (Song et al., 2021b;a), the stable solving of current ODE samplers implicitly requires or assumes that the discretized update operator $T_{\boldsymbol{\theta}}$ is contractive ($\| T_{\boldsymbol{\theta}}(\boldsymbol{x}) - T_{\boldsymbol{\theta}}(\boldsymbol{y})\| \leq L\|\boldsymbol{x} - \boldsymbol{y}\|$ with $L < 1$). This contractivity requirement underlies the design of efficient solvers for inference (Liu et al., 2022; Lu et al., 2022; Zhao et al., 2023; Zhang & Chen, 2023; Zheng et al., 2023b; Lu et al., 2025; Tong et al., 2025).

We identify a fundamental tension overlooked in ODE-based fast inference generation algorithms (DDIM, DPM-Solver, etc.): efficient inference requires large steps and expressive high-Lipschitz models, while convergence demands strong contractivity (Figure 1). Our theoretical analysis reveals that

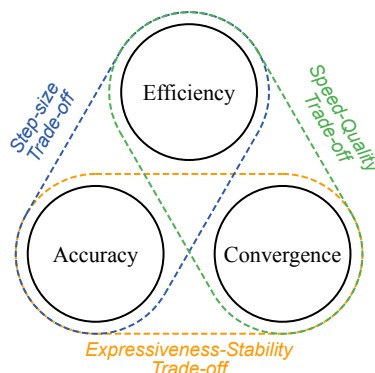

Figure 1: Inference triangle.

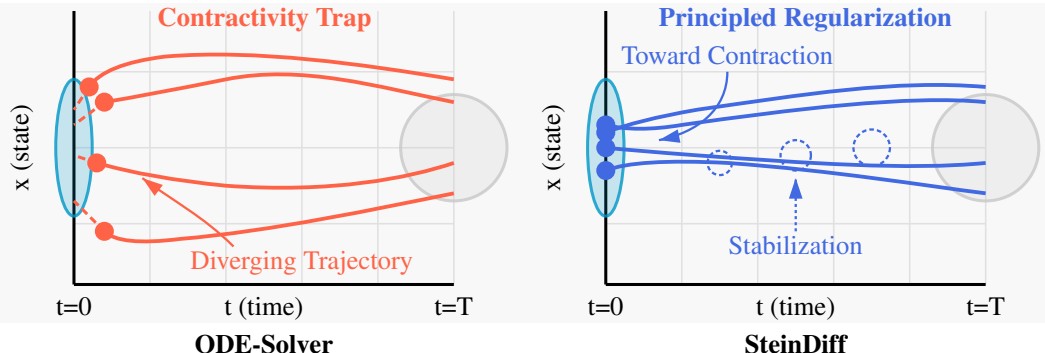

Figure 2: Illustration of denoising trajectories with and without principled stabilization. Efficient ODE solvers fail to satisfy the necessary contractivity constraint ($L_T < 1$) due to large steps and model expressiveness, thereby manifesting compounded error accumulation and trajectory divergence (Left); SteinDiff resolves this critical stability gap by leveraging Stein regularization to impose an adaptive, step-wise correction, ensuring stable convergence to the data manifold (Right).

this unrecognized conflict manifests throughout the denoising trajectory: model expressiveness needed for complex distributions (Raghu et al., 2017) inherently violates the contractivity critical for stable convergence, causing unstable trajectories and degraded sample quality (Figure 2, left). To address this dilemma in the ODE-based diffusion inference, we propose an alternative optimization framework that explicitly relaxes strict contractivity requirements while promoting algorithmic stability and geometric consistency through principled adaptive regularization (Figure 2, right).

In this work, we propose *SteinDiff*, a principled approach that shifts diffusion inference from the implicit contractivity trap to adaptive regularization. Extensive experiments validate the effectiveness of SteinDiff in large-step inference. Our method delivers the following contributions:

- We identify a contractivity trap in efficient inference of DMs and propose a regularization framework for efficient ODE solving, inspired by the Krasnosel'skiĭ-Mann formulation.
- We derive efficient closed-form regularization estimators via Stein's identity without reference guidance, preserving model expressiveness and adapting step-wise to local geometry.
- We not only provide a reference-free solution to bridge a critical stability gap in inference, but also reveal a key principle behind the stability of model parameterizations like EDM by decoupling stability correction into a task driven by a purely geometric signal.

## 2 RELATED WORK

Diffusion models are a class of powerful generative models that excel at generating high-quality samples through denoising refinement, grounded in a well-established theoretical framework (Langevin et al., 1908; Jarzynski, 1997; Neal, 2001; Sohl-Dickstein et al., 2015). Song et al. (Song & Ermon, 2019; 2020) developed modeling techniques using score matching, while Ho et al. (Ho et al., 2020) reformulated DMs into a practical and tractable framework. Song et al. (Song et al., 2021b) unified score-based models and DMs using stochastic differential equations (SDEs), forming the principal framework. However, DMs face the significant challenge of balancing efficiency and quality. Various training and inference methods have recently emerged to address this challenge.

Training-based methods accelerate DMs through post-training or new modeling strategies. Latent DMs (Rombach et al., 2022) boost efficiency by operating in lower-dimensional spaces. Progressive distillation (Salimans & Ho, 2022; Meng et al., 2023) and Consistency Models (CMs) (Song et al., 2023; Luo et al., 2023) enable few-step generation through knowledge distillation and self-consistency learning. Flow matching (FMs) (Lipman et al., 2023; Liu et al., 2023; 2024) optimizes generation paths by learning velocity fields, while EDM (Karras et al., 2022) improves samples using optimized weighting schemes. Shortcut models (Frans et al., 2025) achieve efficiency by step-aware network learning. Additional methods are also explored in (Watson et al., 2022; Wang et al., 2023; Zheng et al., 2023a; Kim et al., 2023; Zhou et al., 2024a; Kim et al., 2024; Wimbauer et al., 2024; Ma et al., 2024b; Zhou et al., 2024b; Karras et al., 2024b; Sauer et al.; Karras et al., 2024a; Zhang et al., 2024; Ma et al., 2024a; Zhang et al., 2025; Tong et al., 2025; Geng et al., 2025).

Inference-focused methods improve DMs by optimizing the inference process without retraining. DDIM (Song et al., 2021a) establishes deterministic sampling using non-Markovian processes. Building on deterministic ODEs, e.g., PNDMs (Liu et al., 2022), denoising solvers have achieved significant progress: DPM-Solver (Lu et al., 2022) uses exponential integrators for acceleration, DEIS (Zhang & Chen, 2023) addresses numerical stiffness, and DPM-Solver++ (Lu et al., 2025) proposes a data-based prediction scheme. UniPC (Zhao et al., 2023) provides a predictor-corrector framework, DPM-Solver-v3 (Zheng et al., 2023b) optimizes speed with pre-computed empirical model statistics, while restart sampling (Xu et al., 2024) refines generation paths with a restart mechanism to mitigate errors. Additional methods include schedule optimizations (Jolicoeur-Martineau et al., 2021; Karras et al., 2022; Xue et al., 2024; Chen et al., 2024; Sabour et al., 2024), discretization techniques (Wizadwongsa & Suwajanakorn, 2023; Guo et al., 2023; Li et al., 2023; Gonzalez et al., 2023; Zhao et al., 2024), and parallel techniques (Shih et al., 2023; Tang et al., 2024).

**Limitations and Our Contribution**   While training-based methods for DMs achieve impressive results, they suffer from expensive training costs, often compromise the iterative refinement advantages that make DMs powerful, and can face GAN-like training instabilities. In contrast, inference-time methods preserve the advantages of DMs without the extra cost or risks. However, these methods face inherently conflicting requirements: efficiency demands large step sizes with high expressiveness (large Lipschitz constants), yet stable convergence requires strong contractivity. We propose SteinDiff, which breaks this deadlock by replacing hard contractivity constraints with adaptive Stein regularization, enabling both stable efficient inference and theoretical convergence guarantees.

## 3   PRELIMINARIES

Diffusion models (Sohl-Dickstein et al., 2015; Ho et al., 2020; Song et al., 2021b; Karras et al., 2022; 2024b) are powerful generative models that learn to reverse a predefined noise-adding process with conditional Gaussian marginals. For efficient generative inference, the reverse process is formulated as the deterministic ODE trajectory (Song et al., 2021b;a; Liu et al., 2022). The ODE governs the trajectory of a sample evolving from pure noise $\boldsymbol{x}_N \sim \mathcal{N}(\boldsymbol{0}, \hat{\sigma}_N^2\mathbf{I})$ to a clean data sample $\boldsymbol{x}_0$. We adopt the unified PF ODE formulation where the score function, $\nabla_{\boldsymbol{x}} \log p_t(\boldsymbol{x}_t)$, is related to a trained noise prediction network $\boldsymbol{\epsilon_\theta}(\boldsymbol{x}_t, t)$ via $-\boldsymbol{\epsilon_\theta}/\sigma_t$. This leads to the following diffusion ODE:

$$\frac{\mathrm{d}\boldsymbol{x}}{\mathrm{d}t} = f(t)\boldsymbol{x}_t + \frac{g^2(t)}{2\sigma_t}\boldsymbol{\epsilon_\theta}(\boldsymbol{x}_t, t), \tag{1}$$

where $f(t)$ and $g(t)$ are schedule-dependent functions and are defined as $f(t) := \frac{\mathrm{d}\log\alpha_t}{\mathrm{d}t}$ and $g^2(t) := \frac{\mathrm{d}\sigma_t^2}{\mathrm{d}t} - 2\frac{\mathrm{d}\log\alpha_t}{\mathrm{d}t}\sigma_t^2$ in variational DMs (Kingma et al., 2021). Rather than relying on the implicit contractivity of the iterative operators used to solve this ODE-based generation process, our work introduces a principled inference framework to guide them towards convergence.

## 4   METHOD

DMs generate high-quality samples by progressively mapping noise into structured data. To analyze the underlying dynamics for inference, we first formulate their updates as an iterative operator.

### 4.1   FROM DIFFUSION ODES TO ITERATIVE OPERATORS

The denoising process can be parameterized by predicting the clean data $\boldsymbol{x}_0$ from a noisy state $\boldsymbol{x}_t$:

$$\boldsymbol{x_\theta}(\boldsymbol{x}_t, t) = \frac{\boldsymbol{x}_t - \sigma_t\boldsymbol{\epsilon_\theta}(\boldsymbol{x}_t, t)}{\alpha_t}. \tag{2}$$

Substituting this parameterization into the diffusion ODE shown in Eq. (1), one has

$$\frac{\mathrm{d}\boldsymbol{x}}{\mathrm{d}t} = \left(f(t) + \frac{g^2(t)}{2\sigma_t^2}\right)\boldsymbol{x}_t - \alpha_t\frac{g^2(t)}{2\sigma_t^2}\boldsymbol{x_\theta}(\boldsymbol{x}_t, t). \tag{3}$$

In practice, one can tailor the inference algorithms for DMs by numerically solving this diffusion ODE using discretization techniques. For any time interval $[t, s]$, the details are provided in Appendix B, we can express the discretized mapping of this ODE from $\boldsymbol{x}_s$ to $\boldsymbol{x}_t$ as the operator $\mathrm{T}_{\boldsymbol{\theta}}(\boldsymbol{x}_s)$:

$$\boldsymbol{x}_t = \mathrm{T}_{\boldsymbol{\theta}}(\boldsymbol{x}_s) := \frac{\sigma_t}{\sigma_s}\boldsymbol{x}_s + \sigma_t\,\mathrm{F}_{\boldsymbol{\theta}}(\boldsymbol{x}_s), \tag{4}$$

where $\mathrm{F}_{\boldsymbol{\theta}}(\boldsymbol{x}_s) := \int_{\kappa(s)}^{\kappa(t)} \boldsymbol{x}_{\boldsymbol{\theta}}\left(\boldsymbol{x}_{\phi(\tau)}, \phi(\tau)\right) \mathrm{d}\tau$ accounts for the model's contribution, $\kappa(t) := \frac{\alpha_t}{\sigma_t}$ and its square represents the signal-to-noise ratio (SNR), and $\phi\left(\kappa(t)\right) := t$ denotes its inverse function.

## 4.2 THE CONTRACTIVITY TRAP: ANALYSIS AND IMPLICATIONS

*The numerical stability of the iterative process, particularly in the finite-step regime, relies on the discretized operators $\mathrm{T}_{\boldsymbol{\theta}}$ being contractive to suppress error accumulation.* Drawing on the contraction mapping principle as a stability criterion framework, we ideally require $\mathrm{T}_{\boldsymbol{\theta}}$ to satisfy

$$\|\mathrm{T}_{\boldsymbol{\theta}}(\boldsymbol{x}) - \mathrm{T}_{\boldsymbol{\theta}}(\boldsymbol{y})\| \leq L_{\mathrm{T}}\|\boldsymbol{x} - \boldsymbol{y}\|, \tag{5}$$

with Lipschitz constant $L_{\mathrm{T}} < 1$. However, this requirement poses a fundamental tension with practical diffusion-generation demands. To demonstrate this, we examine the first-order discretization or DDIM, which serves as the foundation for advanced inference algorithms. Then, we have

$$\mathrm{F}_{\boldsymbol{\theta}}(\boldsymbol{x}_s) = \int_{\kappa(s)}^{\kappa(t)} \boldsymbol{x}_{\boldsymbol{\theta}}\left(\boldsymbol{x}_{\phi(\tau)}, \phi(\tau)\right) \mathrm{d}\tau \approx (\kappa(t) - \kappa(s))\boldsymbol{x}_{\boldsymbol{\theta}}(\boldsymbol{x}_s, s). \tag{6}$$

This yields the discretized inference operator: $\mathrm{T}_{\boldsymbol{\theta}}(\boldsymbol{x}_s) = \frac{\sigma_t}{\sigma_s}\boldsymbol{x}_s + \sigma_t(\kappa(t) - \kappa(s))\boldsymbol{x}_{\boldsymbol{\theta}}(\boldsymbol{x}_s, s)$. Denote $h_t := \kappa(t) - \kappa(s)$. Clearly, $h_t > 0$ for the denoising direction. By rigorous analysis detailed in Appendix C, an upper bound for the Lipschitz constant of this discretized operator is: $L_{\mathrm{T}} \leq \frac{\sigma_t}{\sigma_s} + \sigma_t h_t L_{\boldsymbol{x}_{\boldsymbol{\theta}}}$, where $L_{\boldsymbol{x}_{\boldsymbol{\theta}}}$ denotes the Lipschitz constant of the data prediction function $\boldsymbol{x}_{\boldsymbol{\theta}}(\boldsymbol{x}_t, t)$ with respect to its input $\boldsymbol{x}_t$. For the contractivity demands of $L_{\mathrm{T}} < 1$, we require: $\frac{\sigma_t}{\sigma_s} + \sigma_t h_t L_{\boldsymbol{x}_{\boldsymbol{\theta}}} < 1$, which results in $L_{\boldsymbol{x}_{\boldsymbol{\theta}}}\left(\frac{\alpha_t}{\sigma_t} - \frac{\alpha_s}{\sigma_s}\right) < \frac{1}{\sigma_t} - \frac{1}{\sigma_s}$. However, this creates *an inherent tension triangle* as illustrated in Figure 1: *Efficiency* demands larger step sizes $h_t$ to reduce function evaluations; *Model expressiveness* requires high sensitivity (large $L_{\boldsymbol{x}_{\boldsymbol{\theta}}}$) to capture intricate distributions; and *Stable inference* requires a careful balance between these competing factors to mitigate errors.

**Proposition 4.1 (Violation)** *The update $\mathrm{T}_{\boldsymbol{\theta}}$ violates contractivity when $L_{\boldsymbol{x}_{\boldsymbol{\theta}}} \geq \frac{\sigma_s - \sigma_t}{\sigma_s\alpha_t - \sigma_t\alpha_s}$.*

**Remark 4.2** *Since $\frac{1}{\alpha_t} > \frac{\sigma_s - \sigma_t}{\sigma_s\alpha_t - \sigma_t\alpha_s}$ for noise schedules, models with $L_{\boldsymbol{x}_{\boldsymbol{\theta}}} \geq \frac{1}{\alpha_t}$ will violate even this lenient contractivity bound. Note that $L_{\boldsymbol{x}_{\boldsymbol{\theta}}}$ is influenced by the step size according to Eq. (2).*

**Remark 4.3 (Vulnerability Even in Late-stage)** *The strict contractivity becomes more stringent precisely in the most sensitive phase of generation. Even in the late stage, as $\alpha_t \to 1$, neural network models with sufficient capacity to learn complex data distributions typically exhibit increasing Lipschitz constants for fine-grained discriminative capabilities (Raghu et al., 2017; Bortoli, 2022).*

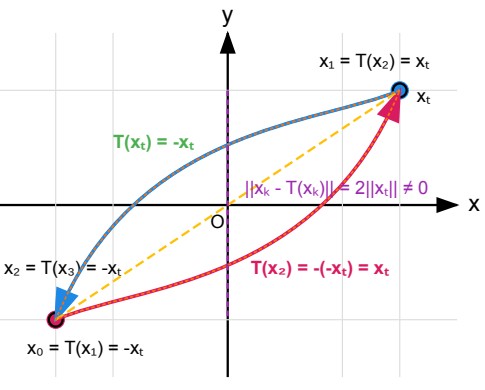

*Practical Consequences.* The violation of contractivity can directly result in error accumulation or degraded quality. To illustrate this, consider the simple yet instructive case where $\mathrm{T} = -\mathrm{I}d$ (the negative identity operator). Starting from any input $\boldsymbol{x}_N \neq \boldsymbol{0}$, the iterations oscillate indefinitely between points of equal magnitude but opposite signs, never converging to the desired solution. This example, while simple, demonstrates how the failure of contractivity can completely derail the inference process, as visualized in Figure 3. We empirically verify this in Figure 4, which confirms that the update operator inherently violates the stability condition ($L_{\mathrm{T}} > 1$) in practice. Since enforcing strict contractivity would severely limit model expressiveness, we instead turn to the following regularization approach.

Figure 3: Oscillation occurs when $\mathrm{T} = -\mathrm{I}d$ ($L_{\mathrm{T}} = 1$). Blue and red arrows depict alternating steps ($k$ vs. $k - 1$) between $\boldsymbol{x}_t$ and $-\boldsymbol{x}_t$.

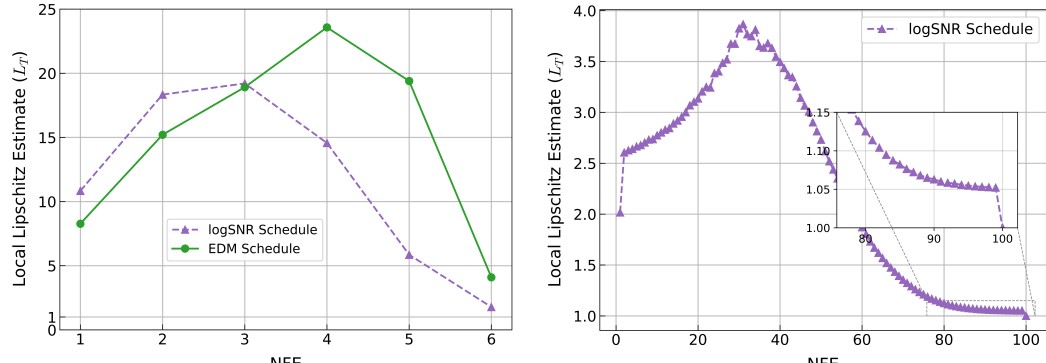

Figure 4: Empirical verification of the contractivity trap for efficient inference. (Left) Violation across schedules (NFE=6) using DPM-Solver++ for EDM2 model. We compare local Lipschitz estimates ($L_T$) for logSNR and EDM schedules. Notably, both schedules significantly violate the stability condition ($L_T < 1$), with peaks reaching $\approx 24$. This confirms that large Lipschitz constants impose a hard limit on the step size for stable inference. (Right) Persistence despite fine-grained steps (NFE=100). Even with small steps, the operator remains expansive ($L_T > 1$) for the majority of the trajectory (see inset). This suggests that resolving the contractivity trap requires explicit regularization of the geometric structure, rather than merely refining the discretization step size.

### 4.3 Towards Regularization via A Krasnosel'skii-Mann Perspective

To address the contractivity trap, we shift to the design framework of regularization methods by drawing on Krasnosel'skii-Mann (KM) theory (Krasnosel'skii, 1955; Groetsch, 1972). Specifically, the KM update can act as a stabilizing mechanism: when $T_{\boldsymbol{\theta}}$ overshoots or oscillates, it provides controlled damping through convex interpolation. Formally, we formulate this update as follows:

$$\boldsymbol{x}_{k-1} = (1 - \gamma_k)\boldsymbol{x}_k + \gamma_k \, T_{\boldsymbol{\theta}}(\boldsymbol{x}_k), \tag{7}$$

where $\gamma_k$ is a relaxation parameter. *This formulation is structurally motivated by classical results on successive approximations (Xu, 2002; Boţ & Csetnek, 2017):*

**Theorem 4.4 (KM Contraction)** *For a nonexpansive operator $T_{\boldsymbol{\theta}}$ (i.e., $\| T_{\theta}(\boldsymbol{x}) - T_{\theta}(\boldsymbol{y})\| \leq \|\boldsymbol{x} - \boldsymbol{y}\|$) in a Hilbert space, the KM iteration (7) converges weakly to a solution of $T_{\boldsymbol{\theta}}$ if $\inf_{k \geq 0}\{\gamma_k\} > 0$.*

*Insight: While $T_{\boldsymbol{\theta}}$ is a discretization operator of time $t$, we leverage the KM structure as a **local stabilization mechanism** at each step $t_k$.* Importantly, this structure mitigates instability while guaranteeing the contractivity that any solution of $T_{\boldsymbol{\theta}}$ remains unchanged (Boţ & Nguyen, 2023). Motivated by this, we adopt this formulation as *a step-wise regularization of the operator update*:

$$\boldsymbol{x}_{k-1} = T_{\boldsymbol{\theta}}(\boldsymbol{x}_k) + (1 - \gamma_k)(\boldsymbol{x}_k - T_{\boldsymbol{\theta}}(\boldsymbol{x}_k)). \tag{8}$$

Herein, the final update consists of the original prediction $T_{\boldsymbol{\theta}}(\boldsymbol{x}_k)$ regularized by a step back towards the current state $\boldsymbol{x}_k$, with the magnitude of this regularization controlled by $(1 - \gamma_k)$.

While better-conditioned parameterizations like $\boldsymbol{v}$-prediction offer improved stability over $\boldsymbol{x}$-prediction (Salimans & Ho, 2022), they do not intrinsically solve the tension between expressiveness and stability in large-step ODE solving. Thus, an explicit regularization framework remains crucial.

### 4.4 Stein's Regularization in the Absence of Target Data

While the KM formulation provides *a structural contraction mechanism for stabilization*, the direct trajectory optimization encounters analytical hurdles arising from the target's functional dependency on the current state. To resolve this, we shift to a population risk minimization framework. Specifically, we identify the target $\boldsymbol{x}^*$ as the latent ground-truth variable $\boldsymbol{x}_0$ underlying the noisy state in DMs. This treats the current state $\boldsymbol{x}_k$ as a sample from the conditional Gaussian $q(\boldsymbol{x}_k|\boldsymbol{x}_0)$, motivating our application of Stein's Identity to derive a closed-form, geometry-aware regularization.

**Theorem 4.5 (Population-Level Regularization)** *Consider the local step-wise KM regularization objective of minimizing the expected distance to the target data:*

$$J(\gamma_k) = \mathbb{E}\big[\|(1 - \gamma_k)\boldsymbol{x}_k + \gamma_k\,\mathrm{T}_{\boldsymbol{\theta}}(\boldsymbol{x}_k) - \boldsymbol{x}^*\|^2\big],$$

*where the target $\boldsymbol{x}^*$ is formulated as the latent ground-truth data variable underlying the noisy state. The expectation is taken over the joint distribution $p(\boldsymbol{x}_k, \boldsymbol{x}^*)$, where $\boldsymbol{x}^* \sim p_0(\boldsymbol{x})$ and $\boldsymbol{x}_k$ follows the transition kernel $p_{t_k}(\boldsymbol{x}_k|\boldsymbol{x}^*)$. Then the minimizer $\gamma_k^*$ satisfies*

$$\gamma_k^* = \frac{\mathbb{E}\langle \boldsymbol{x}_k - \mathrm{T}_{\boldsymbol{\theta}}(\boldsymbol{x}_k),\ \boldsymbol{x}_k - \boldsymbol{x}^*\rangle}{\mathbb{E}\big\|\boldsymbol{x}_k - \mathrm{T}_{\boldsymbol{\theta}}(\boldsymbol{x}_k)\big\|^2}, \tag{9}$$

*where $\langle \cdot, \cdot \rangle$ denotes the inner product. The proof is provided in Appendix D.*

While Theorem 4.5 provides the theoretical optimum, Eq. (9) remains computationally intractable due to the unknown ground truth $\boldsymbol{x}_0$. To address this challenge, we leverage a fundamental property of score-based DMs (Song et al., 2021b; Kingma et al., 2021; Bao et al., 2022; Karras et al., 2022): the reverse-time SDE generates samples whose marginal distributions match the marginals $\boldsymbol{x}_k \sim \mathcal{N}(\alpha_k\boldsymbol{x}_0, \sigma_k^2\mathbf{I})$ of the forward process (Anderson, 1982). Crucially, since the PF ODE preserves these *exact same marginal distributions* $p_t(\boldsymbol{x})$, we can validly invoke Stein's lemma (Stein, 1981; Gorham & Mackey, 2015; Liu & Wang, 2016; Duncan et al., 2023). The validity of Stein's Identity relies solely on the properties of the marginal density $p_k(\boldsymbol{x})$ rather than the specific integration path (SDE vs. ODE). Therefore, we can use Stein's Identity to derive a closed-form, tractable estimator.

**Lemma 4.6 (Stein's Identity)** *For $\boldsymbol{x} \sim \mathcal{N}(\boldsymbol{x}^*, \sigma^2\mathbf{I})$ and any differentiable function $f$ with finite expectations, the following identity holds:*

$$\mathbb{E}\big[f(\boldsymbol{x})(\boldsymbol{x} - \boldsymbol{x}^*)\big] = \sigma^2\mathbb{E}[\nabla f(\boldsymbol{x})]. \tag{10}$$

Now, let $\boldsymbol{u}_k := \boldsymbol{x}_k - \mathrm{T}_{\boldsymbol{\theta}}(\boldsymbol{x}_k)$ denote the residual. The numerator in Eq. (9) can be rewritten as:

$$\mathbb{E}\langle \boldsymbol{u}_k, \boldsymbol{x}_k - \boldsymbol{x}^*\rangle = \mathbb{E}\langle \boldsymbol{u}_k, \boldsymbol{x}_k\rangle - \mathbb{E}\langle \boldsymbol{u}_k, \boldsymbol{x}^*\rangle. \tag{11}$$

Applying this lemma allows us to transform the intractable cross-term in the numerator of $\gamma_k^*$ into a computable divergence term involving the update residual.

**Theorem 4.7 ( Reference-Free Estimation via Stein's Identity )** *Let $\boldsymbol{u}_k := \boldsymbol{x}_k - \mathrm{T}_{\boldsymbol{\theta}}(\boldsymbol{x}_k)$ be the update residual. By exploiting the Gaussian transition kernel $q(\boldsymbol{x}_k|\boldsymbol{x}_0)$ and applying Stein's Identity, the optimal regularization parameter $\gamma_k^*$ admits a closed-form, reference-free expression:*

$$\boxed{\gamma_k^* = \frac{\left(1 - \frac{1}{\alpha_k}\right)\mathbb{E}\langle \boldsymbol{u}_k, \boldsymbol{x}_k\rangle + \frac{\sigma_k^2}{\alpha_k}\mathbb{E}[\nabla \cdot \boldsymbol{u}_k]}{\mathbb{E}\|\boldsymbol{u}_k\|^2}}, \tag{12}$$

*where $\nabla \cdot \boldsymbol{u}_k = \sum_{i=1}^d \partial_{\boldsymbol{x}_k^{(i)}}(\boldsymbol{u}_k)_i$ is the divergence. The detailed proof is provided in Appendix E.*

**Remark 4.8 (Geometric Regularization)** *Unlike standard ODE solvers that purely minimize discretization error, SteinDiff leverages the divergence $\nabla \cdot \boldsymbol{u}_k$ as a geometric detector. This term adaptively rectifies the update in high-curvature regions, effectively acting as a manifold-aware regularizer that prevents the trajectory from overshooting the data manifold.*

### 4.5 STEINDIFF: A STEP-WISE REGULARIZATION FOR EFFICIENT ODE SOLVING

Building on our result in Theorem 4.7, we now propose *SteinDiff*, a class of principled and reference-free algorithms that stabilize efficient ODE solving paths. To implement the theoretical solution in Eq. (12), we approximate the expectations with batch-wise empirical means. Specifically, given a batch of samples of size $B$, we define $\hat{s}_{xu} = \frac{1}{B}\sum\langle \boldsymbol{u}_k, \boldsymbol{x}_k\rangle$ and $\hat{s}_{uu} = \frac{1}{B}\sum\|\boldsymbol{u}_k\|^2$ to approximate the expectations $\mathbb{E}\langle \boldsymbol{u}_k, \boldsymbol{x}_k\rangle$ and $\mathbb{E}\|\boldsymbol{u}_k\|^2$, respectively. For the divergence term $\mathbb{E}[\nabla \cdot \boldsymbol{u}_k]$, we compute $\hat{s}_{div}$ using Hutchinson's estimator: $\hat{s}_{div} = \frac{1}{B}\sum\boldsymbol{v}^\top\nabla_{\boldsymbol{x}}\boldsymbol{u}_k\boldsymbol{v}$, where $\boldsymbol{v} \sim \mathcal{N}(0, \boldsymbol{I})$. Algorithm 1 details the step-wise core procedure utilizing these empirical estimates.

In the following, we analyze and establish the convergence properties of SteinDiff under the practical distribution shift caused by discretization.

---

**Algorithm 1** SteinDiff: Stein's Regularization for Diffusion Model Inference

---

**Require:** State $\boldsymbol{x}_{t_k}$; Model $\boldsymbol{x_\theta}$; Update operator $\mathrm{T}_{\boldsymbol{\theta}}$; Schedule $\{t_k, \alpha_{t_k}, \sigma_{t_k}\}$; Batch size $B$.
1: $\boldsymbol{u}_k \leftarrow \boldsymbol{x}_{t_k} - \mathrm{T}_{\boldsymbol{\theta}}(\boldsymbol{x}_{t_k})$                                                    ▷ Calculate residual
2: $\hat{s}_{xu} \leftarrow \frac{1}{B} \sum \langle \boldsymbol{u}_k, \boldsymbol{x}_{t_k} \rangle; \quad \hat{s}_{uu} \leftarrow \frac{1}{B} \sum \|\boldsymbol{u}_k\|^2$          ▷ Compute terms for $\hat{\gamma}_k$
3: $\hat{s}_{div} \leftarrow \frac{1}{B} \sum \boldsymbol{v}^\top \nabla_{\boldsymbol{x}} \boldsymbol{u}_k \boldsymbol{v}$, with $\boldsymbol{v} \sim \mathcal{N}(0, \boldsymbol{I})$
4: $\hat{\gamma}_k \leftarrow \max \left( \frac{(1-1/\alpha_{t_k})\hat{s}_{xu}+(\sigma_{t_k}^2/\alpha_{t_k})\hat{s}_{div}}{\hat{s}_{uu}}, \epsilon \right)$          ▷ $\epsilon > 0$ follows from Theorem 4.4
5: $\boldsymbol{x}_{t_{k-1}} \leftarrow (1 - \hat{\gamma}_k)\boldsymbol{x}_{t_k} + \hat{\gamma}_k \mathrm{T}_{\boldsymbol{\theta}}(\boldsymbol{x}_{t_k})$          ▷ Perform the regularized update
6: **return** $\boldsymbol{x}_{t_{k-1}}$

---

**Theorem 4.9 (Robustness under Discrete Error)** *Let $p_k$ be the grounded-truth marginal distribution and $\tilde{p}_k$ be the estimated distribution produced by inference. Define the score error as:*

$$\mathcal{S}(\tilde{p}_k, p_k) := \left( \mathbb{E}_{\tilde{p}_k} \left[ \|\nabla \log \tilde{p}_k(\boldsymbol{x}) - \nabla \log p_k(\boldsymbol{x})\|^2 \right] \right)^{1/2}. \tag{13}$$

*Assume $\boldsymbol{u}_k$ is $L_u$-Lipschitz continuous and $\mathbb{E}_{p_k}[|\boldsymbol{u}_k|^2] \geq \delta^2 > 0$. When computing the regularization parameter using $\tilde{p}_k$ instead of $p_k$, the resulting parameter $\tilde{\gamma}_k$ satisfies:*

$$|\tilde{\gamma}_k - \gamma_k^*| \leq \frac{C\sigma_k^2 L_u}{\alpha_k \delta^2} \cdot \mathcal{S}(\tilde{p}_k, p_k), \tag{14}$$

*where $C$ is a constant w.r.t. dimension $d$ and growth properties of $\boldsymbol{u}_k$. This bound reveals that: (i) the error scales with score deviation rather than distribution distance, (ii) the method becomes more robust as $\sigma_k \to 0$ in later denoising stages. The detailed proof is provided in Appendix F.*

The score deviation $\mathcal{S}(\tilde{p}_k, p_k)$ measures how much the actual distribution's score function deviates from the ideal conditional Gaussian case, typically scaling as $\mathcal{O}(\Delta t)$ for discretization step size $\Delta t$. The robustness bound immediately implies the following convergence guarantee:

**Theorem 4.10 (Stability Analysis under Bounded Residuals)** *Even under distribution shift from discretization, Stein's regularization iteration imposes a step-wise contraction that suppresses error accumulation towards the irreducible limit determined by the discretization residual:*

$$\mathbb{E}[\|\boldsymbol{x}_{k-1} - \boldsymbol{x}^*\|^2] \leq (1 - c\tilde{\gamma}_k)\mathbb{E}[\|\boldsymbol{x}_k - \boldsymbol{x}^*\|^2] + \mathcal{O}(\mathcal{S}^2(\tilde{p}_k, p_k)), \tag{15}$$

*where $c > 0$ depends on $\mathrm{T}_\theta$, $\mathcal{S}(\tilde{p}_k, p_k) = \mathcal{O}(\Delta t)$. The detailed proof is provided in Appendix G.*

This convergence guarantee ensures that SteinDiff maintains stability even with practical discretization errors, making it robust for actual inference of diffusion ODEs.

**Remark 4.11 (Adaptive)** *SteinDiff automatically adapts to local geometry: as noise level $\sigma_k$ decreases during denoising, $\gamma_k^*$ decreases proportionally to $\sigma_k^2$, providing stronger stabilization precisely when fine details emerge. This eliminates the need for manual parameter scheduling.*

### 4.6 A New Perspective and Understanding for the EDM Framework

The Stein optimal inference framework serves not only as a step-wise stabilizer but also as a theoretical lens to analyze the diffusion parameterizations. We can use the Stein optimal parameter, $\gamma_k^*$, to reveal a principle behind the renowned stability for EDM (Karras et al., 2022; 2024b): its design achieves superior stability by structurally simplifying the underlying correction problem itself.

The optimal correction signal, represented by the numerator of $\gamma_k^*$ can be conceptually decomposed into two components: a drift correction term, $(1 - \frac{1}{\alpha_t})\mathbb{E}\langle\boldsymbol{u}_t, \boldsymbol{x}_t\rangle$, which addresses errors in the overall data scaling, and a geometric correction term, $\frac{\sigma_t^2}{\alpha_t}\mathbb{E}[\nabla \cdot \boldsymbol{u}_t]$, which corrects for deviations from the local geometry of data manifold. A key feature of the EDM framework is its principled network preconditioning, designed to maintain unit variance for the network's inputs and outputs across all noise levels. A direct consequence of this design is that *the data scaling factor $\alpha_t = 1$.* Applying this design condition to our Stein inference parameter, the drift term structurally vanishes:

$$\left(1 - \frac{1}{\alpha_t}\right)\mathbb{E}\langle\boldsymbol{u}_t, \boldsymbol{x}_t\rangle = \left(1 - \frac{1}{1}\right)\mathbb{E}\langle\boldsymbol{u}_t, \boldsymbol{x}_t\rangle = 0. \tag{16}$$

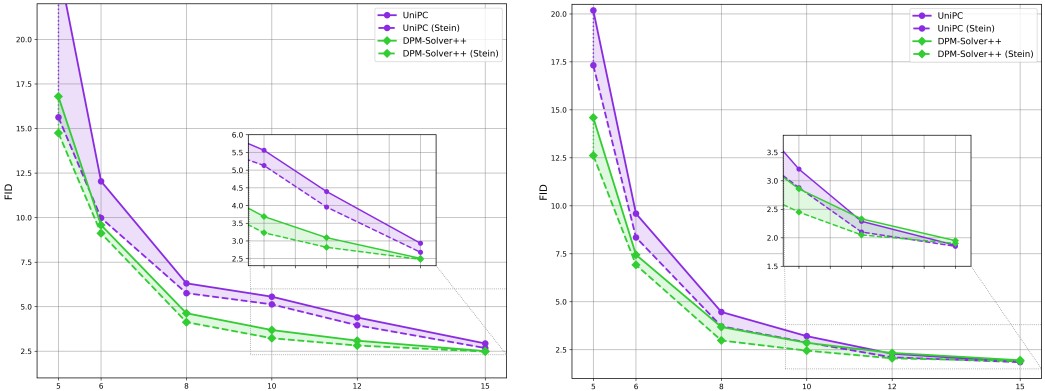

Figure 5: Principled SteinDiff resolves the contractivity trap in few-step inference: at just *3 solver steps* (5 NFE), it stabilizes SOTA solvers like DPM-Solver++ and UniPC (left in each pair) to eliminate severe artifacts and generate high-quality samples (right in each pair) on CIFAR-10 (EDM).

Figure 6: FID ↓ scores for DPM-Solver++ and UniPC using third-order solvers on ImageNet 64×64. The comparison shows performance with and without SteinDiff regularization under both EDM (left) and logSNR (right) noise schedules. The results demonstrate that SteinDiff (dashed lines) consistently improves FID scores across various NFEs for both samplers.

Then, the optimal stability correction required for the EDM framework is purely geometric:

$$\gamma_{k,\text{EDM}}^* \propto \frac{\sigma_t^2}{\alpha_t} \mathbb{E}[\nabla \cdot \boldsymbol{u}_t] = \frac{\sigma_t^2}{1} \mathbb{E}[\nabla \cdot \boldsymbol{u}_t] = \sigma_t^2 \mathbb{E}[\nabla \cdot \boldsymbol{u}_t]. \tag{17}$$

This implies that EDM's stability does not stem from better managing a complex, coupled correction problem, but from *fundamentally simplifying the problem's structure*. By *circumventing the numerical instability* caused by $\alpha_t \to 0$ as $t \to 0$ during the inference process, the challenge is *no longer* to balance algebraic scaling and geometric curvature, but solely to correct for geometric deviations.

By removing the burden of variable scaling, this decoupled preconditioning allows EDM to specialize in learning the geometric structure of the data. A specialized network can learn this task more effectively, leading to an inherently smoother ODE trajectory and a more well-behaved update residual $\boldsymbol{u}_t$. This synergy between static design and simplified dynamic correction is, we argue, the core reason for EDM's renowned stability and performance. Furthermore, our analysis and empirical verification demonstrate that, while distinct from EDM's training framework which optimizes geometric structure to boost model capabilities, SteinDiff explicitly rectifies the trajectory of efficient ODE solvers by capturing the underlying geometric structure during denoising. *This suggests that stable inference depends more on geometric regularization than on fine-grained discretization.*

## 5 EXPERIMENTS

In our experiments, we use the well-established Frechet Inception Distance (FID) and Inception Score (IS) (Heusel et al., 2017; Salimans et al., 2016) metric to measure the quality of the generated images. Furthermore, as the FID score often unfairly favors the models trained with GAN losses and penalizes the diffusion models, we consider an additional metric of FD-DINOv2 (Stein et al., 2023), which replaces the InceptionV3 (Szegedy et al., 2016) encoder of FID by DINOv2 (Oquab et al., 2024) to better align with human perception. Furthermore, we evaluate the efficiency of our

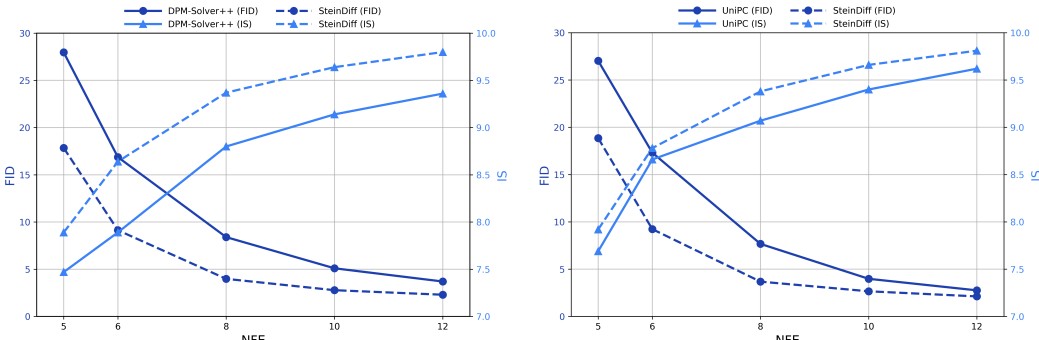

Figure 7: FID ↓ and IS ↑ scores for DPM-Solver++ (left) and UniPC (right) with and without SteinDiff regularization across various NFE. The ablation study demonstrates that SteinDiff (dashed lines) consistently improves both metrics over the baseline solvers on CIFAR-10 with EDM model.

regularized inference scheme using metrics such as the number of solver calls ("Steps") and the Number of Function Evaluations (NFE); lower number metrics indicate higher inference efficiency.

Table 1: Comprehensive performance comparison of various samplers on the imagenet $64 \times 64$ dataset with and without our proposed SteinDiff regularization. The table contrasts the baseline performance (Base) of DPM-Solver++, UniPC, and Heun against the performance with fast Stein-Diff (+Stein) under both logSNR and EDM schedules. Performance is evaluated using FID and FD-DINOv2 metrics, where lower scores are better. The percentage improvement (Improv. %) highlights the consistent and significant gains achieved by SteinDiff across all configurations.

| Metric | Schedule | Steps | DPM-Solver++ | | | UniPC | | | Heun | | |
|---|---|---|---|---|---|---|---|---|---|---|---|
| | | | Base | Stein | Improv. (%) | Base | Stein | Improv. (%) | Base | Stein | Improv. (%) |
| FID | logSNR | 3 | 20.92 | 16.48 | **21.2%** | 27.70 | 19.92 | **28.1%** | 230.05 | 124.69 | **45.8%** |
| | | 3.5 | 12.81 | 10.43 | **18.6%** | 17.04 | 13.05 | **23.4%** | 153.27 | 89.66 | **41.5%** |
| | | 4.5 | 6.16 | 5.22 | **15.3%** | 6.44 | 5.27 | **18.2%** | 51.25 | 33.47 | **34.7%** |
| | | 5.5 | 3.90 | 3.43 | **12.0%** | 3.32 | 2.87 | **13.6%** | 18.60 | 13.40 | **27.9%** |
| | | 6.5 | 2.96 | 2.67 | **9.7%** | 2.45 | 2.21 | **9.8%** | 8.49 | 6.81 | **19.8%** |
| | | 8 | 2.33 | 2.16 | **7.4%** | 2.02 | 1.88 | **7.1%** | 5.12 | 4.42 | **13.6%** |
| | | 10.5 | 1.96 | 1.86 | **5.1%** | 1.80 | 1.71 | **4.9%** | 2.43 | 2.24 | **7.8%** |
| | | 13 | 1.83 | 1.76 | **3.6%** | 1.73 | 1.68 | **2.9%** | 2.08 | 2.02 | **2.7%** |
| FD-DINOv2 | logSNR | 3 | 311.03 | 251.00 | **19.3%** | 425.71 | 315.45 | **25.9%** | 1923.78 | 1035.00 | **46.2%** |
| | | 3.5 | 232.80 | 193.00 | **17.1%** | 331.58 | 258.20 | **22.1%** | 1355.47 | 788.88 | **41.8%** |
| | | 4.5 | 160.59 | 137.79 | **14.2%** | 193.84 | 160.31 | **17.3%** | 697.94 | 452.96 | **35.1%** |
| | | 5.5 | 131.78 | 116.63 | **11.5%** | 138.25 | 120.55 | **12.8%** | 351.30 | 251.88 | **28.3%** |
| | | 6.5 | 118.60 | 107.59 | **9.3%** | 118.69 | 107.90 | **9.1%** | 231.57 | 184.33 | **20.4%** |
| | | 8 | 108.51 | 100.70 | **7.2%** | 106.82 | 99.45 | **6.9%** | 176.95 | 152.02 | **14.1%** |
| | | 10.5 | 101.62 | 95.73 | **5.8%** | 99.91 | 94.71 | **5.2%** | 120.77 | 110.50 | **8.5%** |
| | | 13 | 98.67 | 94.23 | **4.5%** | 97.15 | 93.46 | **3.8%** | 109.78 | 105.17 | **4.2%** |
| FID | EDM | 3 | 17.95 | 14.84 | **17.3%** | 30.02 | 22.58 | **24.8%** | 233.29 | 137.17 | **41.2%** |
| | | 3.5 | 11.00 | 9.45 | **14.1%** | 17.61 | 14.10 | **19.9%** | 81.06 | 50.66 | **37.5%** |
| | | 4.5 | 5.48 | 4.86 | **11.2%** | 7.37 | 6.32 | **14.2%** | 28.66 | 18.91 | **34.0%** |
| | | 5.5 | 3.66 | 3.33 | **9.0%** | 4.37 | 3.88 | **11.3%** | 10.78 | 8.07 | **25.1%** |
| | | 6.5 | 2.87 | 2.67 | **6.9%** | 3.33 | 3.05 | **8.4%** | 5.57 | 4.60 | **17.4%** |
| | | 8 | 2.33 | 2.20 | **5.5%** | 2.59 | 2.42 | **6.5%** | 3.56 | 3.12 | **12.4%** |
| | | 10.5 | 1.97 | 1.89 | **4.1%** | 2.02 | 1.92 | **4.8%** | 2.19 | 2.02 | **7.9%** |
| | | 13 | 1.83 | 1.77 | **3.2%** | 1.81 | 1.74 | **3.8%** | 1.94 | 1.87 | **3.4%** |
| FD-DINOv2 | EDM | 3 | 273.01 | 225.99 | **17.2%** | 449.23 | 335.90 | **25.2%** | 2010.42 | 1178.11 | **41.4%** |
| | | 3.5 | 202.61 | 171.99 | **15.1%** | 314.93 | 250.94 | **20.3%** | 729.42 | 458.81 | **37.1%** |
| | | 4.5 | 142.93 | 125.35 | **12.3%** | 186.25 | 158.68 | **14.8%** | 396.36 | 262.39 | **33.8%** |
| | | 5.5 | 120.09 | 108.68 | **9.5%** | 143.46 | 127.40 | **11.2%** | 252.51 | 188.12 | **25.5%** |
| | | 6.5 | 109.45 | 101.35 | **7.4%** | 124.40 | 114.03 | **8.3%** | 187.93 | 154.29 | **17.9%** |
| | | 8 | 102.53 | 96.28 | **6.1%** | 113.74 | 106.46 | **6.4%** | 153.56 | 134.52 | **12.4%** |
| | | 10.5 | 98.27 | 93.16 | **5.2%** | 104.36 | 99.04 | **5.1%** | 116.79 | 107.21 | **8.2%** |
| | | 13 | 96.57 | 92.42 | **4.3%** | 99.47 | 95.39 | **4.1%** | 108.12 | 102.61 | **5.1%** |

*Variants.* In practice, our reference-free SteinDiff can incorporate a fast scheme from the recent literature (Boţ & Nguyen, 2023) by leveraging the Stein-regularized state, which achieves an improved estimation in the ideal case. Specific details of this variants are provided in Appendix G.1.

*Results.* Our empirical validation confirms the *contractivity trap* and demonstrates that SteinDiff consistently stabilizes efficient ODE solving. On CIFAR-10, SteinDiff eliminates severe artifacts

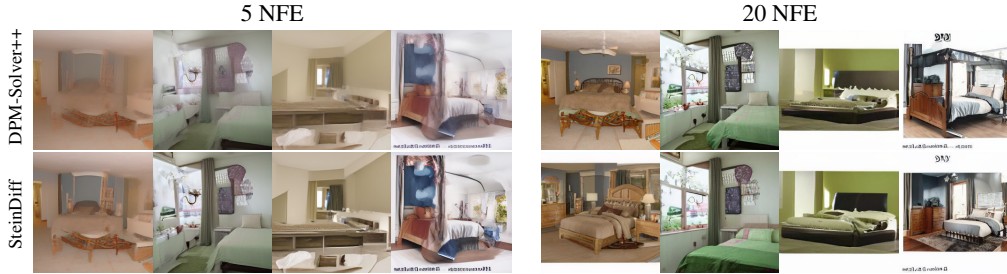

Figure 8: Visual comparison on 256×256 LSUN-Bedrooms. DPM-Solver++ (top) succumbs to the contractivity trap. Conversely, SteinDiff (bottom) resolves this for efficient inference by capturing the underlying geometric structure, improving quality across varying discretization steps.

at just 5 NFE (Figure 5) and significantly boosts image quality across all step sizes. As shown in Figure 7, our method achieves superior FID and IS metrics compared to the baseline, indicating stability against step-size variations. This robustness is further substantiated on the more complex ImageNet 64×64 dataset, where our method delivers substantial gains across various solvers (DPM-Solver++, UniPC, Heun). Notably, these improvements are agnostic to the noise schedule, achieving FID reductions of up to **45.8%** under both EDM and logSNR schedules (Table 1).

Crucially, we also demonstrate the scalability of our approach to Latent DMs. On the LSUN Bedrooms 256×256 dataset, SteinDiff achieves new SOTA performance (Table 2). As visualized in Figure 8, while the baseline DPM-Solver++ succumbs to structural collapse at 5 NFE due to the contractivity trap, SteinDiff preserves high-fidelity geometric structures. This qualitative success confirms that SteinDiff effectively counters the expansive dynamics revealed in Figure 4. Collectively, these results establish our approach as a robust, principled solution for efficient generative inference across diverse modalities. Additional experimental details are provided in Appendix H.

Table 2: Performance comparison on 256×256 LSUN-Bedrooms using a Latent Diffusion model. We evaluate FID scores against DPM-Solver++ across various NFEs, where lower scores indicate better image quality. The results demonstrate that our step-wise regularization method (SteinDiff) significantly outperforms the baseline variants at all tested NFEs.

| Method | Model | NFE | | | | | | |
|---|---|---|---|---|---|---|---|---|
| | | 5 | 6 | 8 | 10 | 12 | 15 | 20 |
| DPM-Solver++ (2m) | Latent Diffusion, LSUN_beds-256 | 21.29 | 10.97 | 5.13 | 3.88 | 3.52 | 3.34 | 3.25 |
| DPM-Solver++ (3m) | | 18.61 | 8.52 | 4.15 | 3.61 | 3.43 | 3.28 | 3.17 |
| SteinDiff (fast) | | **7.64** | **4.71** | **3.72** | **3.38** | **3.01** | **2.86** | **2.77** |

## CONCLUSION

We identify and formalize the *contractivity trap* in diffusion models through both theoretical analysis and empirical validation. To resolve this fundamental problem, we propose *SteinDiff*, a principled regularization framework. At its core, SteinDiff uniquely employs Stein's identity to derive a closed-form, reference-free regularization that explicitly captures the underlying geometric structure of the data manifold during efficient ODE solving. The efficacy of this principled approach is validated by our experiments, demonstrating its ability to maintain structural integrity where traditional methods succumb to instability. Crucially, SteinDiff incurs only marginal overhead, circumventing the need for increased NFEs, model retraining, or expensive schedule optimization, thus offering a decisive advantage over resource-intensive alternatives. Beyond providing step-wise regularization, our analytical approach offers a deeper contribution: a new theoretical lens that rationalizes the inherent stability of EDM, guiding the future co-design of generative models and their inference strategies.

**Limitations and Broader Implications.** While SteinDiff effectively resolves the contractivity trap for efficient ODE solving, it leaves room for future exploration. Crucially, integrating this inference regularization framework with de-biasing and safety mechanisms remains essential; as our correction operates on the sampling trajectory to ensure geometric stability, it is not explicitly designed to rectify inherent data biases or mitigate the misuse risks exacerbated by highly efficient generation. Furthermore, building on our success with LDMs, future work could extend SteinDiff to non-image modalities and multi-modality synthesis, broadening its impact beyond standard benchmarks.

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

APPENDIX

LLM USAGE STATEMENT

During the preparation of this manuscript, we used large language models (LLMs) as writing assistants. Their role was limited to improving the clarity, precision, and grammatical correctness of the text. All core research methods and results were conceived and executed entirely by the human authors through long-term research efforts. The LLMs did not contribute to research ideation or scientific findings. All authors have thoroughly reviewed and edited the manuscript to ensure scientific accuracy and take full responsibility for the final content.

## A NOTATION AND BACKGROUND FOR DIFFUSION MODELS

Denoising diffusion probabilistic models (DDPMs) define a forward process that adds Gaussian noise to data according to a predefined variance schedule $\{\beta_t\}_{t=1}^N$. Although originally formulated in the discrete-time setting, this transition process admits a natural continuous-time generalization (Kingma et al., 2021). In both cases, the conditional distribution of a noisy sample $\boldsymbol{x}_t$ given the clean sample $\boldsymbol{x}_0$ takes the form

$$q(\boldsymbol{x}_t|\boldsymbol{x}_0) := \mathcal{N}(\boldsymbol{x}_t; \alpha_t \boldsymbol{x}_0, \sigma_t^2 \mathbf{I}), \tag{18}$$

where $\alpha_t$ and $\sigma_t$ are deterministic functions of time controlling the signal and noise scales, respectively. In the discrete DDPM formulation, they are related to the variance schedule by $\alpha_t^2 = \prod_{s=1}^t (1 - \beta_s)$ and $\sigma_t^2 = 1 - \alpha_t^2$. For discretized inference, we denote $\alpha_k := \alpha_{t_k}$ and $\sigma_k := \sigma_{t_k}$ for a chosen timestep sequence $t_k$.

To reverse the forward noising dynamics, a neural network $\boldsymbol{\epsilon}_\theta(\boldsymbol{x}_t, t)$ is trained to predict the noise $\boldsymbol{\epsilon}$ added at each timestep. The training objective is simplified to minimizing the mean squared error (MSE):

$$\mathcal{L} = \mathbb{E}_{t, \boldsymbol{x}_0, \boldsymbol{\epsilon}} \left\| \boldsymbol{\epsilon} - \boldsymbol{\epsilon}_\theta(\alpha_t \boldsymbol{x}_0 + \sigma_t \boldsymbol{\epsilon}, t) \right\|^2, \tag{19}$$

where $t$ is uniformly sampled from $\{1, \dots, N\}$, $\boldsymbol{x}_0 \sim q(\boldsymbol{x}_0)$ represents a sample from the data distribution, and $\boldsymbol{\epsilon} \sim \mathcal{N}(\boldsymbol{0}, \mathbf{I})$ is standard Gaussian noise. Samples are generated by iteratively applying the learned reverse transition $p_\theta(\boldsymbol{x}_{t-1}|\boldsymbol{x}_t)$.

This discrete formulation can be unified with continuous-time diffusion models via stochastic differential equations (SDEs) (Song et al., 2021b). Formally, a unified forward process for different schedules (including VP, VE, and sub-VP schedules) is formulated as:

$$\mathrm{d}\boldsymbol{x}_t = f(t)\boldsymbol{x}_t \, \mathrm{d}t + g(t)\mathrm{d}\boldsymbol{w}, \quad \boldsymbol{x}_0 \sim q_0(\boldsymbol{x}_0), \tag{20}$$

where the drift and diffusion coefficients are linked to the noise schedule through $f(t) := \frac{\mathrm{d}\log\alpha_t}{\mathrm{d}t}$ and $g^2(t) := \frac{\mathrm{d}\sigma_t^2}{\mathrm{d}t} - 2\frac{\mathrm{d}\log\alpha_t}{\mathrm{d}t}\sigma_t^2$ (Kingma et al., 2021). The corresponding reverse-time SDEs of these schedules share a unified form:

$$\mathrm{d}\boldsymbol{x} = [f(t)\boldsymbol{x} - g(t)^2 \nabla_{\boldsymbol{x}} \log p_t(\boldsymbol{x})]\mathrm{d}t + g(t)\mathrm{d}\bar{\boldsymbol{w}}, \tag{21}$$

where $\bar{\boldsymbol{w}}$ is a standard Wiener process in the reverse-time direction. Its deterministic counterpart is the probability flow (PF) ordinary differential equation (ODE), which shares the same marginal distributions as the SDE. It is given by:

$$\mathrm{d}\boldsymbol{x} = \left( f(t)\boldsymbol{x} - \frac{1}{2}g(t)^2 \nabla_{\boldsymbol{x}} \log p_t(\boldsymbol{x}) \right) \mathrm{d}t. \tag{22}$$

Under Gaussian modeling assumptions, the score function $\nabla_{\boldsymbol{x}} \log p_t(\boldsymbol{x})$ and the noise predictor $\boldsymbol{\epsilon}_\theta$ are linked by the relation $\boldsymbol{\epsilon}_\theta(\boldsymbol{x}_t, t) = -\sigma_t \nabla_{\boldsymbol{x}} \log p_t(\boldsymbol{x}_t)$ (Song et al., 2021b; Karras et al., 2022). Substituting this relation into Eq. (22) leads to the diffusion ODE form used in our main text (Preliminaries 3):

$$\frac{\mathrm{d}\boldsymbol{x}}{\mathrm{d}t} = f(t)\boldsymbol{x}_t + \frac{g^2(t)}{2\sigma_t}\boldsymbol{\epsilon}_\theta(\boldsymbol{x}_t, t). \tag{23}$$

## B    Further Details for the Iterative Operator 4

By applying the variation-of-constants formula to Eq. (3), we obtain:

$$\boldsymbol{x}_t = e^{f_1(t)} \left( -\int_s^t e^{-f_1(r)} \frac{\alpha_r g^2(r)}{2\sigma_r^2} \boldsymbol{x}_\theta \left( \boldsymbol{x}_r, r \right) \mathrm{d}r + \boldsymbol{x}_s \right) \tag{24}$$

where $f_1(r) := \int_s^r f(z) + \frac{g^2(z)}{2\sigma_z^2} \mathrm{d}z$, $f(t) = \frac{\mathrm{d}\log\alpha_t}{\mathrm{d}t}$ and $g^2(t) = \frac{\mathrm{d}\sigma_t^2}{\mathrm{d}t} - 2\frac{\mathrm{d}\log\alpha_t}{\mathrm{d}t}\sigma_t^2$. Note that $f_1(r)$ can be rewritten as

$$f_1(r) = \int_s^r \frac{\mathrm{d}\log\alpha_\gamma}{\mathrm{d}\gamma} + \frac{1}{2\sigma_\gamma^2}\frac{\mathrm{d}\sigma_\gamma^2}{\mathrm{d}\gamma} - \frac{\mathrm{d}\log\alpha_\gamma}{\mathrm{d}\gamma}\mathrm{d}\gamma = \int_s^r \frac{1}{\sigma_\gamma}\frac{\mathrm{d}\sigma_\gamma}{\mathrm{d}\gamma}\mathrm{d}\gamma = \log\frac{\sigma_r}{\sigma_s}, \tag{25}$$

thus, $e^{f_1(t)} = \frac{\sigma_t}{\sigma_s}$ and $e^{-f_1(r)} = \frac{\sigma_s}{\sigma_r}$. Then, Eq. (24) can be rewritten as

$$\boldsymbol{x}_t = \frac{\sigma_t}{\sigma_s} \left( -\int_s^t \sigma_s \frac{\alpha_r}{\sigma_r} \frac{g^2(r)}{2\sigma_r^2} \boldsymbol{x}_\theta \left( \boldsymbol{x}_r, r \right) \mathrm{d}r + \boldsymbol{x}_s \right). \tag{26}$$

Note that $\frac{g^2(r)}{2\sigma_r^2} = \frac{1}{\sigma_r}\frac{\mathrm{d}\sigma_r}{\mathrm{d}r} - \frac{1}{\alpha_r}\frac{\mathrm{d}\alpha_r}{\mathrm{d}r}$, so $\frac{\alpha_r}{\sigma_r}\frac{g^2(r)}{2\sigma_r^2} = -\frac{\mathrm{d}}{\mathrm{d}r}\left(\frac{\alpha_r}{\sigma_r}\right)$. Therefore, Eq. (26) can be rewritten as:

$$\boldsymbol{x}_t = \frac{\sigma_t}{\sigma_s} \left( \int_s^t \sigma_s \boldsymbol{x}_\theta \left( \boldsymbol{x}_r, r \right) \frac{\mathrm{d}}{\mathrm{d}r}\left(\frac{\alpha_r}{\sigma_r}\right) \mathrm{d}r + \boldsymbol{x}_s \right) = \frac{\sigma_t}{\sigma_s}\boldsymbol{x}_s + \sigma_t \int_s^t \boldsymbol{x}_\theta \left( \boldsymbol{x}_r, r \right) \mathrm{d}\frac{\alpha_r}{\sigma_r}. \tag{27}$$

This result is equivalent to Eq. (4).

## C    The upper bound of $L_\mathrm{T}$

For any two inputs $\boldsymbol{x}_s^{(1)}, \boldsymbol{x}_s^{(2)}$, we have:

$$\mathrm{T}_{\boldsymbol{\theta}}(\boldsymbol{x}_s^{(1)}) - \mathrm{T}_{\boldsymbol{\theta}}(\boldsymbol{x}_s^{(2)}) = \frac{\sigma_t}{\sigma_s}(\boldsymbol{x}_s^{(1)} - \boldsymbol{x}_s^{(2)}) + \sigma_t(\kappa(t) - \kappa(s))[\boldsymbol{x}_{\boldsymbol{\theta}}(\boldsymbol{x}_s^{(1)}, s) - \boldsymbol{x}_{\boldsymbol{\theta}}(\boldsymbol{x}_s^{(2)}, s)] \tag{28}$$

From the parameterization in Eq. (2), we obtain:

$$\boldsymbol{x}_{\boldsymbol{\theta}}(\boldsymbol{x}_s^{(1)}, s) - \boldsymbol{x}_{\boldsymbol{\theta}}(\boldsymbol{x}_s^{(2)}, s) = \frac{1}{\alpha_s}[(\boldsymbol{x}_s^{(1)} - \boldsymbol{x}_s^{(2)}) - \sigma_s(\boldsymbol{\epsilon}_{\boldsymbol{\theta}}(\boldsymbol{x}_s^{(1)}, s) - \boldsymbol{\epsilon}_{\boldsymbol{\theta}}(\boldsymbol{x}_s^{(2)}, s))] \tag{29}$$

Assuming the noise prediction network $\boldsymbol{\epsilon}_{\boldsymbol{\theta}}(\boldsymbol{x}, t)$ satisfies the Lipschitz condition with respect to $\boldsymbol{x}$:

$$\|\boldsymbol{\epsilon}_{\boldsymbol{\theta}}(\boldsymbol{x}^{(1)}, t) - \boldsymbol{\epsilon}_{\boldsymbol{\theta}}(\boldsymbol{x}^{(2)}, t)\| \leq L_\epsilon \|\boldsymbol{x}^{(1)} - \boldsymbol{x}^{(2)}\|, \tag{30}$$

we can bound:

$$\|\boldsymbol{x}_{\boldsymbol{\theta}}(\boldsymbol{x}_s^{(1)}, s) - \boldsymbol{x}_{\boldsymbol{\theta}}(\boldsymbol{x}_s^{(2)}, s)\| \leq \frac{1 + \sigma_s L_\epsilon}{\alpha_s}\|\boldsymbol{x}_s^{(1)} - \boldsymbol{x}_s^{(2)}\| \tag{31}$$

Combining these results, we obtain:

$$\| \mathrm{T}_{\boldsymbol{\theta}}(\boldsymbol{x}_s^{(1)}) - \mathrm{T}_{\boldsymbol{\theta}}(\boldsymbol{x}_s^{(2)})\| \leq \frac{\sigma_t}{\sigma_s}\|\boldsymbol{x}_s^{(1)} - \boldsymbol{x}_s^{(2)}\| + \sigma_t(\kappa(t) - \kappa(s))\frac{1 + \sigma_s L_\epsilon}{\alpha_s}\|\boldsymbol{x}_s^{(1)} - \boldsymbol{x}_s^{(2)}\| \tag{32}$$

$$= \left[ \frac{\sigma_t}{\sigma_s} + \sigma_t(\kappa(t) - \kappa(s))\frac{1 + \sigma_s L_\epsilon}{\alpha_s} \right] \|\boldsymbol{x}_s^{(1)} - \boldsymbol{x}_s^{(2)}\| \tag{33}$$

Recalling that $\kappa(t) = \frac{\alpha_t}{\sigma_t}$, we can simplify:

$$\sigma_t(\kappa(t) - \kappa(s))\frac{1 + \sigma_s L_\epsilon}{\alpha_s} = (1 + \sigma_s L_\epsilon)\left( \frac{\alpha_t}{\alpha_s} - \frac{\sigma_t}{\sigma_s} \right). \tag{34}$$

Therefore, an upper bound of the Lipschitz constant of the discretized iterative operator $\mathrm{T}_{\boldsymbol{\theta}}$ under first-order Euler approximation as follows:

$$L_\mathrm{T} \leq \frac{\sigma_t}{\sigma_s} + (1 + \sigma_s L_\epsilon)\left( \frac{\alpha_t}{\alpha_s} - \frac{\sigma_t}{\sigma_s} \right). \tag{35}$$

This can be equivalently written as:

$$L_\mathrm{T} \leq (1 + \sigma_s L_\epsilon)\frac{\alpha_t}{\alpha_s} - \sigma_s L_\epsilon \frac{\sigma_t}{\sigma_s} \tag{36}$$

The Lipschitz constant $L_\mathrm{T}$ directly depends on the Lipschitz constant $L_\epsilon$ of the noise prediction network and the ratios of the diffusion scheduling parameters $\alpha_t/\alpha_s$ and $\sigma_t/\sigma_s$. When $L_\mathrm{T} < 1$, the operator $\mathrm{T}_{\boldsymbol{\theta}}$ is contractive, ensuring the stability and convergence of the iterative inference process.

## D  PROOF OF THEOREM 4.5 FOR REGULARIZATION PARAMETERS

*Proof.* We expand

$$J(\gamma_k) = \mathbb{E}\big\| (1 - \gamma_k)\boldsymbol{x}_k + \gamma_k \, \mathrm{T}_{\boldsymbol{\theta}}(\boldsymbol{x}_k) - \boldsymbol{x}^* \big\|^2 \tag{37}$$

$$= \mathbb{E}\big\| \boldsymbol{x}_k - \boldsymbol{x}^* - \gamma_k \, (\boldsymbol{x}_k - T_{\boldsymbol{\theta}}(\boldsymbol{x}_k)) \big\|^2 \tag{38}$$

$$= \mathbb{E}\|\boldsymbol{x}_k - \boldsymbol{x}^*\|^2 \; - \; 2\,\gamma_k \, \mathbb{E}\langle \boldsymbol{x}_k - \boldsymbol{x}^*, \, \boldsymbol{x}_k - \mathrm{T}_{\boldsymbol{\theta}}(\boldsymbol{x}_k)\rangle \; + \; \gamma_k^2 \, \mathbb{E}\|\boldsymbol{x}_k - \mathrm{T}_{\boldsymbol{\theta}}(\boldsymbol{x}_k)\|^2. \tag{39}$$

Setting $\partial J/\partial \gamma_k = 0$ yields

$$-2\,\mathbb{E}\langle \boldsymbol{x}_k - \boldsymbol{x}^*, \, \boldsymbol{x}_k - \mathrm{T}_{\boldsymbol{\theta}}(\boldsymbol{x}_k)\rangle + 2\,\gamma_k \, \mathbb{E}\|\boldsymbol{x}_k - \mathrm{T}_{\boldsymbol{\theta}}(\boldsymbol{x}_k)\|^2 = 0, \tag{40}$$

then

$$\gamma_k = \frac{\mathbb{E}\langle \boldsymbol{x}_k - \boldsymbol{x}^*, \, \boldsymbol{x}_k - \mathrm{T}_{\boldsymbol{\theta}}(\boldsymbol{x}_k)\rangle}{\mathbb{E}\|\boldsymbol{x}_k - \mathrm{T}_{\boldsymbol{\theta}}(\boldsymbol{x}_k)\|^2} \tag{41}$$

from which the stated formula follows. $\square$

## E  PROOF OF THEOREM 4.7 FOR STEIN-BASED ESTIMATION

*Proof.* Within the theoretical framework of score-based differential equation modeling, the Gaussian marginal structure is theoretically guaranteed for the reverse process, as established by Anderson's time-reversal theorem (Anderson, 1982) and demonstrated in score-based DMs (Song et al., 2021b). Thus, during the inference process, we have $\boldsymbol{x}_k \sim \mathcal{N}(\alpha_k \boldsymbol{x}^*, \sigma_k^2 \mathbf{I})$, based on the Gaussian transition kernel of the forward process of DMs, where $\boldsymbol{x}^* \sim p_0(\boldsymbol{x})$. Then, applying Stein's identity component-wise to $\boldsymbol{x}_k \sim \mathcal{N}(\alpha_k \boldsymbol{x}^*, \sigma_k^2 \mathbf{I})$, we have

$$\mathbb{E}[(\boldsymbol{u}_k)_i \cdot \alpha_k \boldsymbol{x}^*] = \mathbb{E}[(\boldsymbol{u}_k)_i \boldsymbol{x}_k] - \sigma_k^2 \mathbb{E}[\nabla(\boldsymbol{u}_k)_i]. \tag{42}$$

Summing over all components and rearranging, we have

$$\mathbb{E}\langle \boldsymbol{u}_k, \boldsymbol{x}_k - \alpha_k \boldsymbol{x}^*\rangle = \sigma_k^2 \mathbb{E}[\nabla \cdot \boldsymbol{u}_k], \tag{43}$$

where $\nabla \cdot \boldsymbol{u}_k = \sum_{i=1}^d \partial_{x_k^{(i)}}(\boldsymbol{u}_k)_i$ is the divergence. Therefore:

$$\alpha_k \mathbb{E}\langle \boldsymbol{u}_k, \boldsymbol{x}^*\rangle = \mathbb{E}[\langle \boldsymbol{u}_k, \boldsymbol{x}_k\rangle] - \sigma_k^2 \mathbb{E}[\nabla_{\boldsymbol{x}_k} \cdot \boldsymbol{u}_k]. \tag{44}$$

Substituting the above results into Eq. (11), we have

$$\mathbb{E}\langle \boldsymbol{u}_k, \boldsymbol{x}_k - \boldsymbol{x}^*\rangle = \mathbb{E}\langle \boldsymbol{u}_k, \boldsymbol{x}_k\rangle - \frac{1}{\alpha_k}\big(\mathbb{E}\langle \boldsymbol{u}_k, \boldsymbol{x}_k\rangle - \sigma_k^2 \mathbb{E}[\nabla \cdot \boldsymbol{u}_k]\big) \tag{45}$$

$$= \Big(1 - \frac{1}{\alpha_k}\Big) \mathbb{E}\langle \boldsymbol{u}_k, \boldsymbol{x}_k\rangle + \frac{\sigma_k^2}{\alpha_k} \mathbb{E}[\nabla \cdot \boldsymbol{u}_k]. \tag{46}$$

The proof is complete by substituting this result into Eq. (9). $\square$

## F  PROOF OF THEOREM 4.9 FOR ROBUSTNESS UNDER DISCRETE ERROR

We provide the complete proof of the robustness bound under distribution shift.

### F.1  SETUP AND NOTATION

Let $p_k(\boldsymbol{x}) = \mathcal{N}(\alpha_k \boldsymbol{x}^*, \sigma_k^2 \boldsymbol{I})$ be the theoretical Gaussian marginal and $\tilde{p}_k$ be the actual distribution. Recall:

$$\gamma_k^* = \frac{(1 - \frac{1}{\alpha_k})\mathbb{E}_{p_k}[\langle \boldsymbol{u}_k, \boldsymbol{x}_k\rangle] + \frac{\sigma_k^2}{\alpha_k}\mathbb{E}_{p_k}[\nabla \cdot \boldsymbol{u}_k]}{\mathbb{E}_{p_k}[|\boldsymbol{u}_k|^2]}, \tag{47}$$

$$\tilde{\gamma}_k = \frac{(1 - \frac{1}{\alpha_k})\mathbb{E}_{\tilde{p}_k}[\langle \boldsymbol{u}_k, \boldsymbol{x}_k\rangle] + \frac{\sigma_k^2}{\alpha_k}\mathbb{E}_{\tilde{p}_k}[\nabla \cdot \boldsymbol{u}_k]}{\mathbb{E}_{\tilde{p}_k}[|\boldsymbol{u}_k|^2]}. \tag{48}$$

## F.2 Key Lemmas

**Lemma F.1 (Stein's Identity for Gaussians)** *For a Gaussian distribution $p_k = \mathcal{N}(\alpha_k \boldsymbol{x}^*, \sigma_k^2 \boldsymbol{I})$ and any differentiable vector field $\boldsymbol{f} : \mathbb{R}^d \to \mathbb{R}^d$ with suitable growth conditions, then:*

$$\mathbb{E}_{p_k}[\langle \boldsymbol{f}(\boldsymbol{x}), \boldsymbol{x} - \alpha_k \boldsymbol{x}^* \rangle] = \sigma_k^2 \mathbb{E}_{p_k}[\nabla \cdot \boldsymbol{f}(\boldsymbol{x})]. \tag{49}$$

*Proof.* For the $i$-th component:

$$\mathbb{E}_{p_k}[f_i(\boldsymbol{x})(x_j - \alpha_k x_j^*)] = \int f_i(\boldsymbol{x})(x_j - \alpha_k x_j^*) p_k(\boldsymbol{x}) d\boldsymbol{x} \tag{50}$$

$$= -\sigma_k^2 \int f_i(\boldsymbol{x}) \frac{\partial \log p_k}{\partial x_j} p_k(\boldsymbol{x}) d\boldsymbol{x} \tag{51}$$

$$= \sigma_k^2 \int \frac{\partial f_i}{\partial x_j} p_k(\boldsymbol{x}) d\boldsymbol{x} = \sigma_k^2 \mathbb{E}_{p_k} \left[ \frac{\partial f_i}{\partial x_j} \right]. \tag{52}$$

Summing over $j = i$ gives the result. $\square$

**Lemma F.2 (Score Deviation Error)** *Let $p_k = \mathcal{N}(\alpha_k \boldsymbol{x}^*, \sigma_k^2 \boldsymbol{I})$ be the ideal Gaussian marginal distribution and $\tilde{p}_k$ be the actual distribution produced by inference. For any Lipschitz continuous vector field $\boldsymbol{f} : \mathbb{R}^d \to \mathbb{R}^d$ with suitable growth conditions to ensure the expectations are finite, the following inequality holds:*

$$\left| \mathbb{E}_{\tilde{p}_k}[\langle \boldsymbol{f}(\boldsymbol{x}), \boldsymbol{x} - \alpha_k \boldsymbol{x}^* \rangle] - \sigma_k^2 \mathbb{E}_{\tilde{p}_k}[\nabla \cdot \boldsymbol{f}(\boldsymbol{x})] \right| \leq \sigma_k^2 \sqrt{\mathbb{E}_{\tilde{p}_k}[\|\boldsymbol{f}(\boldsymbol{x})\|^2]} \cdot \mathcal{S}(\tilde{p}_k, p_k). \tag{53}$$

*Proof.* Using integration by parts:

$$\mathbb{E}_{\tilde{p}_k}[\boldsymbol{f} \cdot (\boldsymbol{x} - \alpha_k \boldsymbol{x}^*)] - \sigma_k^2 \mathbb{E}_{\tilde{p}_k}[\nabla \cdot \boldsymbol{f}] \tag{54}$$

$$= \mathbb{E}_{\tilde{p}_k}[\boldsymbol{f} \cdot (\boldsymbol{x} - \alpha_k \boldsymbol{x}^*)] + \sigma_k^2 \mathbb{E}_{\tilde{p}_k}[\boldsymbol{f} \cdot \nabla \log \tilde{p}_k] \tag{55}$$

$$= \mathbb{E}_{\tilde{p}_k}[\boldsymbol{f} \cdot (\boldsymbol{x} - \alpha_k \boldsymbol{x}^* + \sigma_k^2 \nabla \log \tilde{p}_k)]. \tag{56}$$

For Gaussian $p_k$, we have $\boldsymbol{x} - \alpha_k \boldsymbol{x}^* = -\sigma_k^2 \nabla \log p_k(\boldsymbol{x})$. Therefore:

$$\mathbb{E}_{\tilde{p}_k}[\boldsymbol{f} \cdot (\boldsymbol{x} - \alpha_k \boldsymbol{x}^*)] - \sigma_k^2 \mathbb{E}_{\tilde{p}_k}[\nabla \cdot \boldsymbol{f}] = \sigma_k^2 \mathbb{E}_{\tilde{p}_k}[\boldsymbol{f} \cdot (\nabla \log \tilde{p}_k - \nabla \log p_k)]. \tag{57}$$

Applying the Cauchy-Schwarz inequality, we complete the proof. $\square$

## F.3 Main Proof of Theorem 4.9

We begin by decomposing the error between the estimated parameter $\tilde{\gamma}_k$ and the optimal parameter $\gamma_k^*$. This error can be expressed as

$$\tilde{\gamma}_k - \gamma_k^* = \frac{N_{\tilde{p}} - N_p \cdot R}{D_{\tilde{p}}}, \tag{58}$$

where:

$$N_p = (1 - \tfrac{1}{\alpha_k}) \mathbb{E}_{p_k}[\langle \boldsymbol{u}_k, \boldsymbol{x}_k \rangle] + \tfrac{\sigma_k^2}{\alpha_k} \mathbb{E}_{p_k}[\nabla \cdot \boldsymbol{u}_k], \tag{59}$$

$$N_{\tilde{p}} = (1 - \tfrac{1}{\alpha_k}) \mathbb{E}_{\tilde{p}_k}[\langle \boldsymbol{u}_k, \boldsymbol{x}_k \rangle] + \tfrac{\sigma_k^2}{\alpha_k} \mathbb{E}_{\tilde{p}_k}[\nabla \cdot \boldsymbol{u}_k], \tag{60}$$

$$D_{\tilde{p}} = \mathbb{E}_{\tilde{p}_k}[|\boldsymbol{u}_k|^2], \quad R = D_{\tilde{p}} / \delta^2. \tag{61}$$

The numerator $N_p$ of the optimal parameter can be simplified. Under the ideal Gaussian distribution $p_k$, we apply Stein's identity as shown in Lemma F.1, which yields:

$$N_p = (1 - \tfrac{1}{\alpha_k}) \mathbb{E}_{p_k}[\langle \boldsymbol{u}_k, \boldsymbol{x}_k \rangle] + \tfrac{\sigma_k^2}{\alpha_k} \mathbb{E}_{p_k}[\nabla \cdot \boldsymbol{u}_k] \tag{62}$$

$$= (1 - \tfrac{1}{\alpha_k}) \left( \mathbb{E}_{p_k}[\langle \boldsymbol{u}_k, \alpha_k \boldsymbol{x}^* \rangle] + \sigma_k^2 \mathbb{E}_{p_k}[\nabla \cdot \boldsymbol{u}_k] \right) + \tfrac{\sigma_k^2}{\alpha_k} \mathbb{E}_{p_k}[\nabla \cdot \boldsymbol{u}_k] \tag{63}$$

$$= (1 - \tfrac{1}{\alpha_k}) \alpha_k \mathbb{E}_{p_k}[\langle \boldsymbol{u}_k, \boldsymbol{x}^* \rangle] + \sigma_k^2 \mathbb{E}_{p_k}[\nabla \cdot \boldsymbol{u}_k]. \tag{64}$$

The core of the analysis lies in bounding the term $N_{\tilde{p}} - N_p \cdot R$. The key difference arises from the failure of Stein's identity for $\tilde{p}_k$. By Lemma F.2:

$$|N_{\tilde{p}} - N_p \cdot R| \tag{65}$$

$$\leq \left|(1 - \tfrac{1}{\alpha_k})\left(\mathbb{E}_{\tilde{p}_k}[\langle \boldsymbol{u}_k, \boldsymbol{x}_k\rangle] - \tfrac{\sigma_k^2}{\alpha_k}\mathbb{E}_{\tilde{p}_k}[\nabla \cdot \boldsymbol{u}_k]\right)\right| + |N_p||R - 1| \tag{66}$$

$$\leq (1 - \tfrac{1}{\alpha_k})\sigma_k^2\sqrt{\mathbb{E}_{\tilde{p}_k}[|\boldsymbol{u}_k|^2]} \cdot \mathcal{S}(\tilde{p}_k, p_k) + |N_p||R - 1|. \tag{67}$$

Furthermore, under the Lipschitz assumption on $\boldsymbol{u}_k$, the variation in the denominator can also be controlled by the score deviation:

$$|D_{\tilde{p}} - \delta^2| \leq C_1 \cdot W_2(\tilde{p}_k, p_k) \leq C_2 \cdot \mathcal{S}(\tilde{p}_k, p_k), \tag{68}$$

where the second inequality uses the relationship between the Wasserstein distance and score deviation. Combining these bounds for both the numerator and denominator, and for a sufficiently small score deviation such that $D_{\tilde{p}} \geq \delta^2/2$, we arrive at the final result. Since $\boldsymbol{u}_k$ is $L_u$-Lipschitz and $D_{\tilde{p}} = O(\delta^2)$, we obtain:

$$|\tilde{\gamma}_k - \gamma_k^*| \leq \frac{(1 - \tfrac{1}{\alpha_k})\sigma_k^2\sqrt{D_{\tilde{p}}} \cdot \mathcal{S}(\tilde{p}_k, p_k) + |N_p| \cdot O(\mathcal{S}(\tilde{p}_k, p_k))}{D_{\tilde{p}}} \tag{69}$$

$$\leq \frac{2\sigma_k^2}{\alpha_k\delta^2} \cdot \mathcal{S}(\tilde{p}_k, p_k) \cdot \left(\sqrt{D_{\tilde{p}}} + O(1)\right). \tag{70}$$

Since $\boldsymbol{u}_k$ is $L_u$-Lipschitz and $D_{\tilde{p}} = O(\delta^2)$, we obtain:

$$|\tilde{\gamma}_k - \gamma_k^*| \leq \frac{C\sigma_k^2 L_u}{\alpha_k\delta^2} \cdot \mathcal{S}(\tilde{p}_k, p_k), \tag{71}$$

where $C$ is a constant that absorbs dimension-dependent terms and growth properties of $\boldsymbol{u}_k$.

### F.4 Additional Remarks

**Remark F.3** *The constant $C$ can be made explicit as $C = 2(1 + \sqrt{d} + M_u/\delta)$ where $M_u$ bounds the growth of $\boldsymbol{u}_k$.*

**Remark F.4** *The score deviation $\mathcal{S}(\tilde{p}_k, p_k)$ can be bounded by the discretization step size: $\mathcal{S}(\tilde{p}_k, p_k) = O(\Delta t)$ under standard regularity conditions.*

**Remark F.5** *The bound becomes tighter as $\sigma_k \to 0$, confirming that SteinDiff is most robust in the critical late stages of denoising where fine details emerge.*

## G Proof of Theorem 4.10 for Stability Analysis under Bounded Residuals

*Proof.* To analyze the stability of SteinDiff, we examine the evolution of the expected squared error relative to the latent ground truth $\boldsymbol{x}_0$, rather than a deterministic ODE trajectory. Let $\boldsymbol{e}_k = \boldsymbol{x}_k - \boldsymbol{x}^*$ denote the error at step $k$, where $\boldsymbol{x}^* \sim q_{data}(x)$. The update rule is given by:

$$\boldsymbol{x}_{k-1} = (1 - \tilde{\gamma}_k)\boldsymbol{x}_k + \tilde{\gamma}_k T_\theta(\boldsymbol{x}_k), \tag{72}$$

can then be expressed in terms of this error. By subtracting $\boldsymbol{x}^*$ from both sides, we obtain:

$$\boldsymbol{e}_{k-1} = \boldsymbol{x}_{k-1} - \boldsymbol{x}^* = (1 - \tilde{\gamma}_k)(\boldsymbol{e}_k + \boldsymbol{x}^*) + \tilde{\gamma}_k T_\theta(\boldsymbol{x}_k) - \boldsymbol{x}^* \tag{73}$$

$$= (1 - \tilde{\gamma}_k)\boldsymbol{e}_k + \tilde{\gamma}_k(T_\theta(\boldsymbol{x}_k) - \boldsymbol{x}^*). \tag{74}$$

To facilitate the analysis of its squared norm, we further decompose the term $(T_\theta(\boldsymbol{x}_k) - \boldsymbol{x}^*)$ by adding and subtracting $T_\theta(\boldsymbol{x}^*)$, which yields the key expression for the one-step error evolution:

$$\boldsymbol{e}_{k-1} = \underbrace{(1 - \tilde{\gamma}_k)\boldsymbol{e}_k + \tilde{\gamma}_k(T_\theta(\boldsymbol{x}_k) - T_\theta(\boldsymbol{x}^*))}_{\text{Contraction Component}} + \underbrace{\tilde{\gamma}_k(T_\theta(\boldsymbol{x}^*) - \boldsymbol{x}^*)}_{\text{Local Consistency Error}}. \tag{75}$$

Taking the squared norm of this expression, $\|\boldsymbol{e}_{k-1}\|^2$, results in a sum of the squared norms of these two components plus their inner product. We now bound each of these resulting terms.

First, we analyze the contraction component. A standard and powerful result for Krasnosel'skii-Mann iterations on nonexpansive operators, such as the ODE discretization operator $T_\theta$, provides a fundamental inequality. While a general nonexpansive operator has a Lipschitz constant $L = 1$, discrete ODE solvers typically exhibit stronger local contractivity ($L < 1$) near the ground-truth. This allows us to state that for some constant $c > 0$:

$$\|(1 - \tilde{\gamma}_k)\boldsymbol{e}_k + \tilde{\gamma}_k(T_\theta(\boldsymbol{x}_k) - T_\theta(\boldsymbol{x}^*))\|^2 \le (1 - c\tilde{\gamma}_k)\|\boldsymbol{e}_k\|^2. \tag{76}$$

This term ensures that, in the absence of other errors, the iteration is contractive and converges.

Next, we address the terms involving the local consistency error, which arise because $\boldsymbol{x}^*$ is not solution of the discrete operator $T_\theta$. The cross-term from expanding equation 75 can be bounded using the Cauchy-Schwarz inequality:

$$2\tilde{\gamma}_k|\langle(1 - \tilde{\gamma}_k)\boldsymbol{e}_k + \tilde{\gamma}_k(T_\theta(\boldsymbol{x}_k) - T_\theta(\boldsymbol{x}^*)), T_\theta(\boldsymbol{x}^*) - \boldsymbol{x}^*\rangle| \tag{77}$$
$$\le 2\tilde{\gamma}_k\|(1 - \tilde{\gamma}_k)\boldsymbol{e}_k + \tilde{\gamma}_k(T_\theta(\boldsymbol{x}_k) - T_\theta(\boldsymbol{x}^*))\| \cdot \|T_\theta(\boldsymbol{x}^*) - \boldsymbol{x}^*\| \tag{78}$$
$$\le 2\tilde{\gamma}_k\|\boldsymbol{e}_k\| \cdot \|T_\theta(\boldsymbol{x}^*) - \boldsymbol{x}^*\|, \tag{79}$$

where the last step uses the contractive property established above. The crucial observation that connects this proof to our main robustness result (Theorem 4.9) is that the magnitude of this fixed-point error is directly controlled by the score deviation. The analysis in Theorem 4.9 implies that both the parameter error $|\tilde{\gamma}_k - \gamma_k^*|$ and the local consistency error deviation are consequences of the distribution shift, leading to:

$$\mathbb{E}_{\tilde{p}_k}[\|T_\theta(\boldsymbol{x}^*) - \boldsymbol{x}^*\|^2] = O(\mathcal{S}^2(\tilde{p}_k, p_k)). \tag{80}$$

Finally, we combine these bounds by taking the expectation of $\|\boldsymbol{e}_{k-1}\|^2$ over the distribution $\tilde{p}_k$. The expectation of the full expansion becomes:

$$\mathbb{E}[\|\boldsymbol{e}_{k-1}\|^2] \le (1 - c\tilde{\gamma}_k)\mathbb{E}[\|\boldsymbol{e}_k\|^2] + 2\tilde{\gamma}_k\mathbb{E}[\|\boldsymbol{e}_k\| \cdot \|T_\theta(\boldsymbol{x}^*) - \boldsymbol{x}^*\|] + \tilde{\gamma}_k^2\mathbb{E}[\|T_\theta(\boldsymbol{x}^*) - \boldsymbol{x}^*\|^2]. \tag{81}$$

Applying Young's inequality to the cross-term ($2ab \le a^2 + b^2$) and using the fact that $\tilde{\gamma}_k \in (0, 1]$, we find that the terms involving the local consistency error are absorbed into a single additive term of order $O(\mathcal{S}^2(\tilde{p}_k, p_k))$. This establishes the final suboptimality bound for the iteration:

$$\mathbb{E}[\|\boldsymbol{x}_{k-1} - \boldsymbol{x}^*\|^2] \le (1 - c\tilde{\gamma}_k)\mathbb{E}[\|\boldsymbol{x}_k - \boldsymbol{x}^*\|^2] + O(\mathcal{S}^2(\tilde{p}_k, p_k)), \tag{82}$$

thus completing the proof. □

---

**Algorithm 2** SteinDiff with Self-Consistency Correction

**Require:** Model $\boldsymbol{x}_\theta$, Operator $T_\theta$, $\{t_i, \alpha_t, \sigma_t, B\}$.
1: Initialize $\{\boldsymbol{x}_{t_k}^{(i)}, T_\theta(\boldsymbol{x}_{t_k}^{(i)}), t_k, t_{k-1}\}_{i=1}^B$
2: $(\boldsymbol{x}_p^{(i)}, \hat{\gamma}_k) \leftarrow \text{SteinDiff}(\boldsymbol{x}_{t_k}^{(i)}, t_k, t_{k-1}, \boldsymbol{x}_\theta, \{\alpha, \sigma\})$
3: $\boldsymbol{x}_{two}^{(i)} \leftarrow \text{TwoStep}(\boldsymbol{x}_{t_k}^{(i)}, t_k, t_{k-2})$
4: $w = \log(\sigma_{t_{k-1}}/\sigma_{t_k})/\log(\sigma_{t_{k-2}}/\sigma_{t_k})$
5: $\boldsymbol{x}_{alt}^{(i)} \leftarrow (1 - w) \cdot \boldsymbol{x}_{t_k}^{(i)} + w \cdot \boldsymbol{x}_{two}^{(i)}$
6: $\rho_k \leftarrow \exp(-\|\boldsymbol{x}_p^{(i)} - \boldsymbol{x}_{alt}^{(i)}\|^2/(2\sigma_{k-1}^2))$
7: $\boldsymbol{x}_{t_{k-1}}^{(i)} \leftarrow \rho_k \cdot \boldsymbol{x}_p^{(i)} + (1 - \rho_k) \cdot \boldsymbol{x}_{alt}^{(i)}$
8: **return** $\{\boldsymbol{x}_{t_{k-1}}^{(i)}\}_{i=1}^B$

---

### G.1 FAST VARIANTS AND SELF-CONSISTENCY CORRECTION OF STEINDIFF

As we discussed, the KM formulation not only ensures convergence in inference time without any reference solution guidance but also preserves the expressiveness of the vanilla DMs. In this section,

we introduce a fast KM scheme from (Sabach & Shtern, 2017; Boţ & Nguyen, 2023) with improved rates. Specifically, for $\alpha > 2$, $\boldsymbol{x}_0, \boldsymbol{x}_1 \in \mathcal{H}$, and $0 < s \leq 1$, the iteration for $k \geq 1$ is:

$$\boldsymbol{x}_{k-1} := \left(1 - \frac{s\alpha}{2\eta}\right)\boldsymbol{x}_k + \frac{s\alpha}{2\eta}\mathrm{T}_{\boldsymbol{\theta}}(\boldsymbol{x}_k) + \frac{(1-s)\,k}{\eta}(\boldsymbol{x}_k - \boldsymbol{x}_{k+1}) + \frac{sk}{\eta}(\mathrm{T}_{\boldsymbol{\theta}}(\boldsymbol{x}_k) - \mathrm{T}_{\boldsymbol{\theta}}(\boldsymbol{x}_{k+1})), \quad (83)$$

where $\eta = k + \alpha$. This fast KM scheme incorporates more information to achieve better contraction. The contractivity of this fast KM version is guaranteed by the recent result in (Boţ & Nguyen, 2023):

**Theorem G.1 (Contractivity of Fast KM Scheme)** *For a nonexpansive operator $\mathrm{T}_{\boldsymbol{\theta}}$ in a Hilbert space, the fast KM scheme governed by Eq. (83) converges weakly to a solution of $\mathrm{T}_{\boldsymbol{\theta}}$ and exhibits a convergence rate for the consistency residual of $\mathcal{O}(k^{-1})$, i.e.,*

$$\|\boldsymbol{x}_k - \boldsymbol{x}_{k-1}\| = \mathcal{O}(k^{-1}) \quad or \quad \|\boldsymbol{x}_{k-1} - \mathrm{T}(\boldsymbol{x}_{k-1})\| = \mathcal{O}(k^{-1}).$$

Moreover, we also provide a self-consistency correction of SteinDiff in Algorithm 2, which can be used for future information to further mitigate discretization errors caused by the update operator.

## H EXPERIMENTAL SETUP AND DETAILS

Our experiments were conducted to validate the effectiveness of SteinDiff in enhancing state-of-the-art ODE solvers, including DPM-Solver++, UniPC, and the Heun method, using their default public implementations. We performed evaluations on several standard benchmarks with corresponding pre-trained models: a CIFAR-10 EDM model, an ImageNet 64x64 model under both EDM and logSNR noise schedules, and a Latent Diffusion Model for LSUN Bedrooms 256x256. Specifically, experiments on CIFAR-10 and ImageNet were conducted using an NVIDIA 3090 GPU, while evaluations on the LSUN Bedrooms 256x256 dataset utilized an NVIDIA 4090D GPU. Performance was measured using standard metrics, including Fréchet Inception Distance (FID) and Inception Score (IS), with FID scores computed over 50,000 generated samples. We also reported FD-DINOv2, which replaces the standard InceptionV3 encoder with a DINOv2 backbone for a more perceptually aligned evaluation. The SteinDiff regularizer was implemented as detailed in Algorithm 1, utilizing the Hutchinson trace estimator with five random vectors to approximate the divergence term. For the main quantitative results, we employed a fast KM variant to ensure accelerated convergence while maintaining theoretical guarantees. A small safeguard constant, $\epsilon$, was used to ensure numerical stability during the computation of the regularization parameter $\hat{\gamma}_k$.

*Efficient Implementation.* The main computational cost lies in evaluating the divergence term $\nabla \cdot \boldsymbol{u}_k$ in Eq. (12), which naively requires $\mathcal{O}(d)$ vector-Jacobian products. We address this using the Hutchinson trace estimator (Hutchinson, 1989). This reduces the computational cost to a single vector-Jacobian product per sample. The estimator is unbiased and has been widely adopted in modern deep learning applications (Grathwohl et al., 2019; Tsitsulin et al., 2020; Meyer et al., 2021).

*Computational Overhead.* The cost of computing the divergence for a batch of size $B$ using $m$ Hutchinson samples is approximately $m$ times the cost of a single vector-Jacobian product (VJP). In practice, setting $m = 5$, we observe that the overhead remains manageable compared to the neural network evaluation in $\mathrm{T}_{\boldsymbol{\theta}}$. It is important to note that Stein regularization can be deployed adaptively, inserted at specific timesteps where geometric correction is most required, without being applied at every timestep. Crucially, due to this sparse activation, the amortized computational cost of our method is significantly lower than the cumulative cost of increasing solver sampling steps. This ensures that SteinDiff remains more time-efficient than high-NFE baselines. Therefore, although SteinDiff incurs a marginal overhead, it effectively circumvents the need for increased NFEs, model retraining, or expensive schedule optimization, thereby offering a superior efficiency trade-off.

### H.1 DETAILS

We implement `SteinDiff` as a standalone PyTorch module compatible with standard ODE solvers (e.g., `DPM_Solver`). The design strictly adheres to the theoretical derivations in Section 4, enforcing a post-hoc correction step to stabilize the sampling trajectory. The regularization mechanism acts via the `apply` method. Rather than accepting the raw solver output `x_t` as the next state, we enforce a KM formulation update using adaptive interpolation:

$$x_{\text{new}} = (1 - \gamma) \cdot x + \gamma \cdot x_t \quad (84)$$

Here, `x` is the current state, `x_t` is the candidate prediction from the ODE solver, and $\gamma$ is the adaptive regularization parameter computed dynamically at each step. The module intercepts the solver's candidate state and the original state to output the final, stabilized result.

The numerical evaluation of the optimal regularization parameter $\gamma$, following the closed-form derivation in Eq. (12), is implemented within the `compute_optimal_gamma` method. The process begins by defining the update residual $u = x - x_t$ and aggregating the batch-wise statistics, specifically the inner product $\langle u, x \rangle$ (`s_xu`) and the squared norm $\|u\|^2$ (`s_uu`). A critical step involves estimating the divergence term $\nabla \cdot u$. To this end, we provide two strategies: an analytical approximation, which efficiently computes divergence solely on the linear component of the ODE operator but ignores non-linearities; and a full divergence estimator, which utilizes the Hutchinson trace method to capture the full residual divergence. The latter employs `torch.autograd.functional.jvp` combined with antithetic sampling and Rademacher distributions for variance reduction. Finally, these statistics are combined via Stein's Identity to solve for $\gamma$, with the result clamped to a lower bound (typically `1e-6`) to satisfy the convergence constraints of Theorem 4.4.

To balance theoretical precision with computational cost, the module supports three configuration modes controlled via initialization flags. The full mode estimates the full residual divergence (including non-linearities) via the Hutchinson trace estimator at every step, maximizing theoretical rigor at the cost of additional overhead. Conversely, the analytical-only mode relies exclusively on the linear approximation of the ODE operator, adding negligible cost to the inference process. To bridge these extremes, we design an adaptive mode that dynamically switches between estimators based on the noise level $\sigma_t$. This hybrid approach utilizes the fast analytical approximation during early, high-noise stages and transitions to the precise Hutchinson estimator in the critical low-noise regime where fine-grained stability is paramount.

## H.2 MORE EXPERIMENTS

We present further qualitative results demonstrating SteinDiff's robustness across three critical regimes. Figure 10 illustrates the severity of the contractivity trap at a strict budget of 5 NFE. Under these conditions, standard solvers (UniPC, DPM-Solver++) suffer geometric collapse and significant artifacts. SteinDiff (+DPM) effectively counters this drift by regularizing updates towards the data manifold, preserving coherent object structures across diverse classes.

Even when baselines converge at higher steps (10 NFE), geometric regularization yields tangible perceptual benefits. As shown in Figure 11, while DPM-Solver++ produces globally coherent samples, it often smooths over textures. SteinDiff recovers fine-grained details specifically by sharpening features such as animal fur and flower petals, which forces the generation trajectory to adhere more closely to the underlying data geometry.

Regarding scalability, Figure 12 validates our Hutchinson-based trace estimation in high-dimensional spaces. On the LSUN-Bedrooms $256 \times 256$ benchmark (LDM), SteinDiff achieves a SOTA FID of 2.77 at 20 NFE. These results confirm that the proposed approximation remains computationally effective and robust in latent spaces.

Finally, the sensitivity analysis in Figure 9 reveals high robustness to the Hutchinson probe count $m$. Even a single probe ($m = 1$) yields significant gains over the baseline, and performance saturates rapidly at $m = 5$, indicating that our scalar regularization parameter is inherently insensitive to gradient estimation noise.

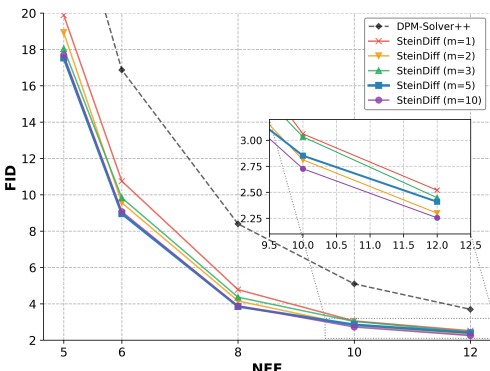

Figure 9: Sensitivity analysis of the Hutchinson probe count ($m$) on CIFAR-10. We compare SteinDiff with varying probe counts $m \in \{1, 2, 3, 5, 10\}$ against the DPM-Solver++ baseline. The results demonstrate that SteinDiff is highly robust to estimation noise, significantly outperforming the baseline even with a single probe ($m = 1$), and performance saturates rapidly at $m = 5$.

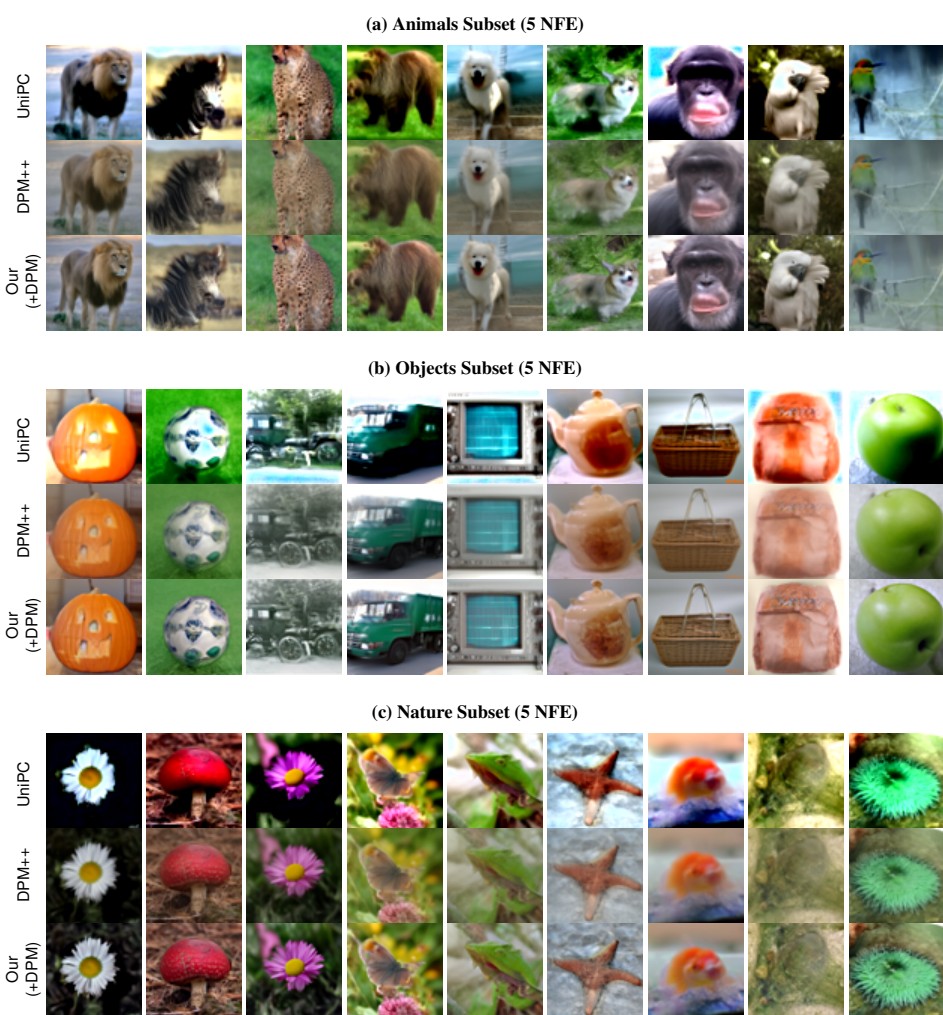

Figure 10: Overcoming the contractivity trap at extreme sparsity (5 NFE). While Efficient solvers like UniPC and DPM-Solver++ (DPM++) suffer from severe structural collapse and artifacts due to insufficient contractivity at large steps, SteinDiff (Our+DPM) successfully stabilizes the inference trajectory of efficient ODE solving. By explicitly correcting the geometric drift, our method preserves semantic fidelity across diverse categories (Animals, Objects, Nature).

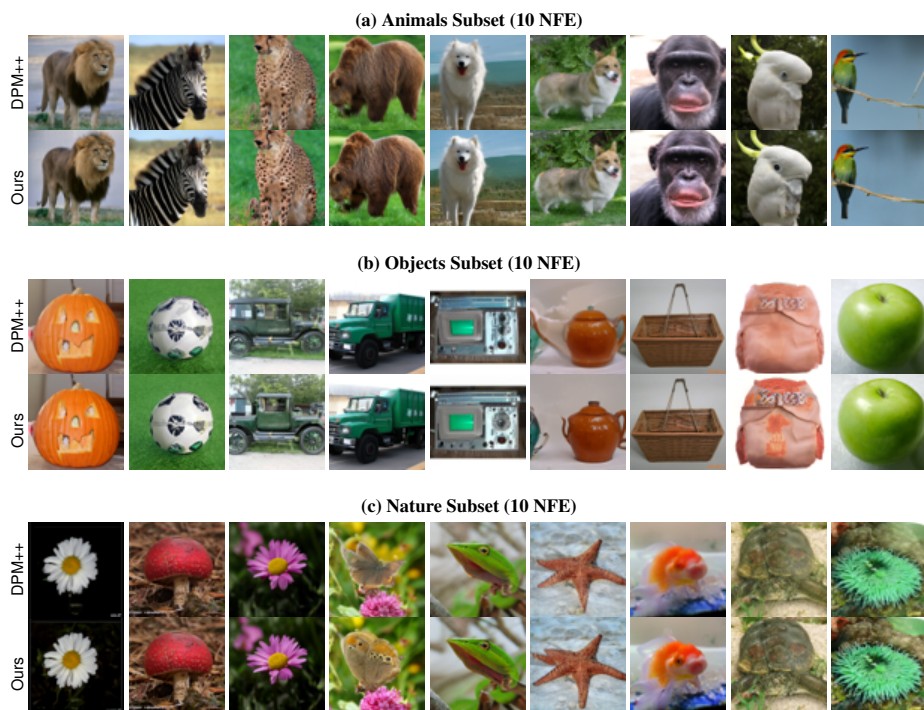

Figure 11: Enhanced fine-grained detail reconstruction at 10 NFE. Comparison between the baseline DPM-Solver++ (DPM++) and our SteinDiff-regularized version (Our). Even when the baseline achieves convergence, SteinDiff significantly refines high-frequency textures (e.g., animal fur, flower petals) and sharpens object boundaries. This demonstrates that our reference-free Stein regularization improves perceptual quality by adhering closer to the data manifold.

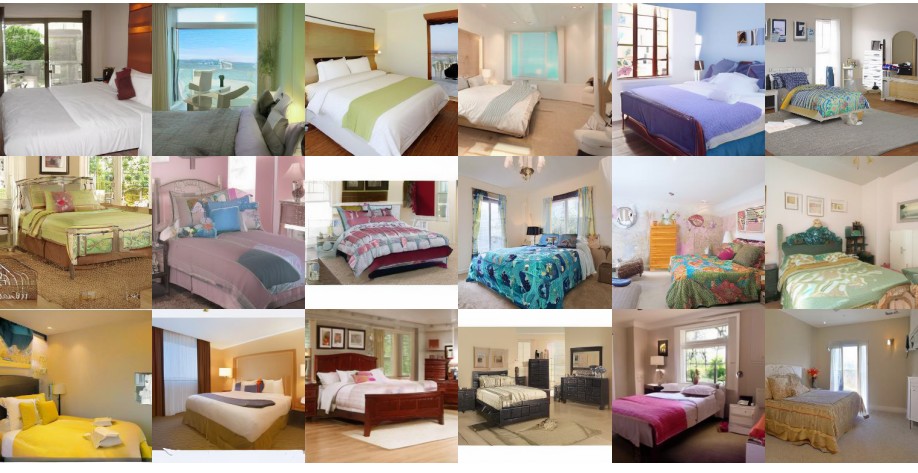

Figure 12: Samples generated by using SteinDiff for efficient DPM-Solver++ solving on LSUN Bedroom at 20 NFE. Achieving a SOTA FID of 2.77, these results empirically validate that our Hutchinson-based trace estimation remains robust and effective in high-dimensional latent spaces, effectively countering concerns regarding scalability and approximation errors.

