# OpenReview forum: "SteinDiff: Resolving the Contractivity Trap via Reference-free Stein Regularization"
_ICLR.cc/2026/Conference — Submitted to ICLR 2026_

### Official Review · Reviewer_iXw4 · 2025-10-19

**Soundness:** 3
**Presentation:** 3
**Contribution:** 3
**Rating:** 6
**Confidence:** 3

**Summary:**

The paper addresses a fundamental limitation in diffusion model inference known as the contractivity trap. This trap arises from the tension between efficiency (large integration steps), expressiveness (complex score networks), and stability (the need for contractive updates in ODE-based sampling). While contractivity is a sufficient condition for convergence, this assumption fails for expressive modern diffusion models, leading to instability and degraded sample quality at low step counts. To overcome this, they propose SteinDiff, a new inference algorithm that enforces convergence guarantees leveraging Krasnosel’skiĭ–Mann fixed-point theory. Specifically, the update rule is re-expressed as a regularization of the standard ODE step with parameters being computed in closed form by leveraging the Stein Identity.  Experiments on standard benchmarks (CIFAR-10, ImageNet 64×64) show consistent improvements in FID and FD-DINOv2 scores under low-NFE conditions, with modest computational overhead.

**Strengths:**

(1) Replacing the Banch theorem with Krasnosel’skiĭ–Mann to enable larger step sizes and improved convergence speed is novel and well motivated
(2) Deriving a closed-form adaptive regularization parameters by leveraging the Stein Identity is really nice
(3) The experiments show meaningful gains (e.g., FID improvements) in low-NFE (number of function evaluations) regimes on standard benchmarks such as CIFAR-10 and ImageNet.

**Weaknesses:**

(1) Motivation and Illustration: To strengthen the motivation, consider adding plots that illustrate how a large Lipschitz constant constrains the step size on Cifar10 or ImageNet. Such visualizations would clarify the “contractivity trap” and emphasize why small steps are necessary for stability.
(2) Experimental Scope: Most experiments are limited to CIFAR-10 and ImageNet-64×64. These datasets are relatively small; extending evaluations to higher-resolution or larger-scale settings would significantly reinforce the paper’s claims about generality and scalability.
(3) Scalability and Clarity: The closed-form expression for γ (Eq. 12) depends on the trace approximation of \nabla u_k, estimated via the Hutchinson method (Algorithm 1). This approach may face scalability issues in high dimensions since only a limited number of probe vectors 𝑣 are used. Moreover, several symbols in Algorithm 1 appear without prior definition in the text—these should be clearly introduced for readability.
(4) Formatting: conclusion should show on page 9

**Questions:**

(1) Could the authors provide plots illustrating that current diffusion models exhibit large Lipschitz constants restricting the allowable step size and leading to instability?

(2) Given the limited scalability of the \gamma solution due to the trace approximation, wouldn't this approach be more suited for latent diffusuion models? Can you show results on large scale datasets? When does the approach break because of this approximation?

---

> ### Author Response · Authors · 2025-11-24
>
> Thank you for your recognition and the insightful feedback on our work. We are particularly grateful for your suggestions. Our responses are summarized as follows.
>
> > W1: Visualization of the "Contractivity Trap" (Response to Weakness 1 & Q1)
>
> Thank you for your valuable suggestions. We have added a new *Figure 4 (Page 5)* to empirically verify the Lipschitz constants of the update operator on CIFAR-10 using the EDM2 model.
>
> *These empirical results further validate our key insights:*
>
> - *Severe Violation of Contractivity:* As shown in *Figure 4 (Left)*, the local Lipschitz estimate ($L_T$) consistently violates the stability condition ($L_T < 1$) across the inference trajectory. Specifically, we observe that $L_T$ peaks at approximately **24**, which is an order of magnitude larger than the stability threshold. This quantitatively confirms that the update operator is highly expansive, leading to error accumulation in standard solvers.
>
> - *Independence from Schedule:* We compared both logSNR and EDM schedules and found that both exhibit this expansive behavior, reinforcing that this is a fundamental property of the learned diffusion ODEs.
>
> - *Persistence at Small Steps:* Furthermore, *Figure 4 (Right)* demonstrates that even with a fine-grained step size ($NFE=100$), the operator remains expansive ($L_T > 1$) for the majority of the trajectory.
>
>
> These empirical results provide strong justification for our proposed Stein regularization, which is designed explicitly to handle such high-Lipschitz operators without forcing a hard constraint that would limit expressiveness.
>
> > W2: Scalability and Experiments on LDMs (Response to Weakness 2 & Q2-Part1)
>
> To address the concern regarding experimental scope and scalability, we have extended our evaluation to the *LSUN-Bedrooms 256x256* dataset using a Latent Diffusion Model (Stable Diffusion). The results from these experiments strongly validate the method's scalability:
>
> - *SOTA Performance (Table 2, Page 10):* We added *Table 2* in the revised manuscript, which compares SteinDiff against DPM-Solver++ on LSUN-Bedrooms. SteinDiff consistently achieves lower FID scores across all NFEs, achieving a new state-of-the-art FID of **2.77** at 20 NFE.
>
> - *Preventing Structural Collapse (Figure 8, Page 10):* We added *Figure 8* to visualize the generated samples at an extremely low budget of *5 NFE*. The results are striking: while the baseline DPM-Solver++ suffers from severe structural collapse (producing incoherent images), SteinDiff successfully regularizes the trajectory, preserving high-fidelity geometric structures and realistic details.
>
> - *Robustness in Latent Space:* These results empirically confirm that our Hutchinson-based trace estimation scales effectively to the high-dimensional latent spaces of LDMs, resolving the concern about the approximation breaking down.
>
>
> By demonstrating success on this higher-resolution, large-scale dataset, we have substantiated the generality of SteinDiff beyond small-scale benchmarks.

---

> ### Author Response · Authors · 2025-11-24
>
> > W3: Robustness of Trace Approximation (Response to Weakness 3 & Q2-Part2)
>
> In the revised version, we performed a rigorous sensitivity analysis on the number of Hutchinson probe vectors ($m$) and validated scalability in high-dimensional latent spaces. Specifically, we observe the following key behaviors:
>
> *Evidence of Robustness and Scalability:*
>
> - *Robustness to Approximation Noise (Figure 9, Appendix H.2):* We tested SteinDiff with varying probe counts $m \in \{1, 2, 3, 5, 10\}$ on CIFAR-10. The results in *Figure 9* are compelling:
>
>   - *Even with a single probe ($m=1$):* SteinDiff significantly outperforms the DPM-Solver++ baseline (e.g., reducing FID from >20 to ~18 at 5 NFE).
>
>   - *Rapid Saturation:* Performance saturates at just *$m=5$*, with negligible gains from further increasing $m$.
>
>   - *Reasoning:* This robustness stems from the fact that we are estimating a *scalar* regularization parameter $\gamma$. The estimation noise averages out effectively, making the method highly resilient even when the gradient estimate is noisy.
>
> - *Scalability to High Dimensions (Figure 12, Appendix H.2):* As demonstrated in our LDM experiments (*Figure 12*), the estimator remains effective in the high-dimensional latent space of Stable Diffusion (LSUN-Bedrooms 256x256), achieving a SOTA FID of *2.77*.
>
>
> The approach does not "break" easily. It remains effective even under the most extreme approximation ($m=1$) and scales successfully to high-dimensional generative tasks.
>
> > W4: Clarity and Formatting (Response to Weakness 3 & 4)
>
> Thank you for the careful reading. We have addressed these points to improve readability and structure.
>
> - We have revised *Algorithm 1 (Page 7)* to explicitly define all symbols. Specifically, we added a comment clarifying that the stabilization parameter $\epsilon > 0$ follows from the convergence constraints of Theorem 4.4. All other variables (e.g., $\hat{s}_{xu}$, $\hat{s}_{div}$) are explicitly defined in Section 4.5 (Page 6, line 318-321) preceding the algorithm.
> - *Formatting and Page Length (Weakness 4):* The reviewer noted that *the conclusion should appear on page 9*. However, to fully address the reviewer's requests for *motivation plots (Figure 4)*, *LDM experiments (Table 2, Figure 8)*, and *sensitivity analysis (Figure 9)*, the revised manuscript has naturally extended to *10 pages* (excluding references). The Conclusion now appears on *Page 10*. We believe this slight extension is justified by the significant inclusion of empirical evidence that strengthens the paper's core claims.
>
> Thanks again for your great effort in reviewing our submission. If our response addresses your concerns, could you please consider updating your assessment and raising the score? If there are still concerns we may have missed, please let us know.

---

### Official Review · Reviewer_XG3j · 2025-10-23

**Soundness:** 3
**Presentation:** 3
**Contribution:** 3
**Rating:** 8
**Confidence:** 4

**Summary:**

The paper attributes inference errors in diffusion models to the contractive assumption (Lipschitz constant < 1) imposed on the discretization operator. This assumption arises from constructing the operator via the Banach–Picard theorem. To address this limitation, the authors propose a regularized discretizer based on Krasnosel’skii–Mann theory, which removes the need for contraction. However, the regularized formulation introduces the challenge of selecting an optimal interpolation weight. By casting this as an optimization problem, the authors show that the optimal weight admits a closed-form solution when applying Stein’s identity. The resulting algorithm is termed SteinDiff.

**Strengths:**

- The paper presents a rigorous and well-structured critique of the so-called contractivity trap in DM inference.
 - The adoption of the Krasnosel’skii–Mann (KM) framework provides a theoretically sound justification for why the approach by Karras et al. (2022) performs well.
 - The paper offers comprehensive error analysis, convergence results, and compelling empirical evidence supporting the proposed method.

Overall, I found the paper technically solid and enjoyable to read.

**Weaknesses:**

It is not clear that relaxing the operator from contractive ($L<1$) (from Banach–Picard theorem) to nonexpansive ($L \leq 1$) in Theorem 4.4 is sufficient to accommodate the Lipschitz constant of the data predictive function. Further clarification or justification of this transition would strengthen the argument.

#### Minor Comments

- Several symbols are undefined, including $\alpha$ and $\sigma_t$ (and by extension $\sigma_k$ and $\sigma_s$). A notation section in the appendix would fix this if space is a problem.
- The term non-expansive mapping is not formally defined.
- The acronym DM is first introduced in Section 2 but used earlier in Section 1.
- Figures 2 and 3 require more descriptive captions, as they are difficult to interpret based on the current text.
- Line 215 is missing a "to" as in "we shift to the design" and a "the" before contractive trap.

**Questions:**

* Why does the relaxation from $L < 1$ to $L \leq 1$ suffice to guarantee convergence of DM inference under the update of Eq. (7) with $γ^*$ from Eq. (12)?

---

> ### Author Response · Authors · 2025-11-24
>
> Thank you for your insightful and constructive review, which helps us improve the quality of our paper. We are glad that you recognize our contribution and find our work is *technically solid and enjoyable to read*.
>
> We have carefully revised the manuscript to address the reviewer’s single (and very astute) point of clarification, as well as all minor comments.
>
> > 1. Is relaxing to $L \le 1$ sufficient?
>
> Thank you for pointing out this concern. We fully agree with the reviewer's intuition: if we strictly adhered to classical fixed-point theory to find a global fixed point, merely relaxing to $L \le 1$ would indeed be insufficient for high-Lipschitz networks.
>
> To resolve this, we have carefully revised Section 4.3 (renamed to *Towards Regularization via Krasnosel'skii-Mann Perspective*) and Section 4.4 to explicitly distinguish between our theoretical inspiration and the practical algorithmic mechanism:
>
> 1. *KM Theory as a Perspective*:
>   In the revised Section 4.3, we clarify KM theory as a *design framework* rather than a rigid constraint. We clarify that the KM update form acts as a *stabilizing mechanism* to provide controlled damping. We explicitly state in line 256 that the theoretical fixed-point guarantee holds *in the ideal case.*
>
> 2. *Step-wise Regularization*: For clarity, we have re-expressed the formulation in *Section 4.3*, specifically stating: *In practice, we adopt and re-express this KM formulation as a step-wise regularization of the standard ODE update (Eq. 8)*.
>   This clarifies that we do not rely on the operator satisfying $L \le 1$ globally.
>
> 3. *Handling High-Lipschitz via Stein's Identity*: Crucially, instead of relying on the operator bound $L \le 1$, we pivot to a direct optimization objective in Section 4.4. We utilize Stein's Identity to compute the optimal $\gamma_k^*$ that minimizes the expected error at each step.  By capturing the local geometry of the data manifold, SteinDiff effectively adjusts each inference step to better align with the true data distribution, leading to improved stability and convergence during the denoising process.
>
> 4. *Robustness Verification*: Our convergence proof (Theorem 4.9) further supports this by demonstrating that the algorithm minimizes error bounded by the score deviation $\mathcal{S}(\tilde{p}_k, p_k)$, rather than relying on the strict Lipschitz constant of the operator.
>
>
> > 2. Minor Comments
>
> Thank you for your detailed suggestions. We have incorporated all of them into the revised version:
>
> - *Notation Section:* We have added a new *Appendix A: Notation and Background* that formally defines $\alpha, \sigma, \alpha_t, \sigma_t$ and their relationships.
>
> - *Definition of Non-expansive:* We have added the formal mathematical definition *inline within Theorem 4.4* (i.e., $\|T(x)-T(y)\| <= \|x-y\|$) for immediate clarity.
>
> - *Acronym DM:* We have ensured that "Diffusion Models (DMs)" is defined upon its first appearance in *Section 1* to avoid confusion.
>
> - *Figure Captions:* We have significantly expanded the captions for *Figures 2 and 3* to explicitly explain the contractivity trap and SteinDiff's stabilization mechanism.
>
> - *Typo:* We have corrected the sentence to "shift to the design" as pointed out.
>
>
>
>
> Thank you again for your thoughtful and constructive review. If our response addresses your concerns, we would be grateful for your continued support. If there are still concerns we may have missed, please let us know.

---

> > ### Comment · Reviewer_XG3j · 2025-11-25
> >
> > With the authors’ clarifications, it is now clear why the non-expansive mapping is a local rather than a global property. I have reviewed the revised Sections 4.3 and 4.4, and they are consistent with the explanation provided in the rebuttal. This addresses my main concern about the paper.
> >
> > I would recommend that the authors further revise the caption for Figure 3. As it stands, it is still difficult to infer the logic behind the arrows and the color scheme, and how these elements relate to the corresponding operators, even when referring to the main text.

---

> > > ### Author Response · Authors · 2025-11-25
> > >
> > > Thank you for your assessment and for confirming that our clarifications and revisions regarding the local non-expansive property have successfully addressed your main concern.
> > >
> > > Regarding *Figure 3*, we fully agree that the visual logic needs to be explicit. We will revise the caption to clearly explain the oscillation mechanism depicted (the case of ``T = -I$d$"). Specifically, we will clarify that:
> > >
> > > - *The Arrows* represent the operator's mapping direction, illustrating the indefinite switching between a point $x$ and its symmetric opposite $-x$.
> > >
> > > - *The Colors* (e.g., blue and red) are used to distinguish the alternating steps of this cycle (Step $k$ vs. Step $k+1$), highlighting the non-convergent loop behavior.
> > >
> > >
> > > Thank you again for your detailed review and constructive guidance, which have significantly strengthened our paper.

---

### Official Review · Reviewer_ubxb · 2025-10-31

**Soundness:** 1
**Presentation:** 2
**Contribution:** 2
**Rating:** 2
**Confidence:** 3

**Summary:**

Stable convergence for ODE flow methods during inference/generation for diffusion models typically requires taking small enough step sizes so that the maps are contractive, which increases computation time. The authors refer to this as the "contractivity trap". Can one speed up inference without in spite of this? The authors propose SteinDiff, which allows fast inference without inheriting the problems of non-contractive maps, by interpolating with the previous iterate (as suggested by Krasnosel’ski˘ı-Mann theory). Practically, the authors provide a way to compute the optimal parameters for this regularization from data using Stein's identity. Experiments on show that it improves image generation in the low NFE (Number of Function Evaluations) regime on CIFAR-10 and ImageNet.

**Strengths:**

Speeding up diffusion models while preserving generation quality is an important question. The algorithm is easy to implement, and is a purely inference-time modification. It shows improvements on experiments in a low NFE regime.

**Weaknesses:**

In general, the conceptual explanations/discussions are loose and don't seem supported by theory. In particular, it is not clear to me how the mathematical theory presented (Theorem 4.4 based on KM theory) justifies the actual algorithm, which calls the method into question. The connection between the theory and the actual algorithm is missing. See my key question below.

**Questions:**

The update operator $T_\theta$ implicitly depends on the start and end time $s$ and $t$ (which is unfortunately suppressed in the notation), so during inference, a sequence of these operators which are *different* are composed with each other. However, the theorems about convergence to a fixed point relies on a *fixed* map $T$, so it is unclear to me how they apply. This is a major confusion for me. Please detail carefully in which parts you are considering different $T$'s, and which part you are considering the same $T$, because it is not valid to apply fixed-point theory when the map is changing.

It is unclear what the authors mean by "expressiveness", and especially by the claim that higher expressiveness requires large Lipschitz constant in the T map. This appears to me to be a misunderstanding resulting from conflating a step of discretization (where we want the added update to have Lipschitz constant <1) with the entire map itself (which can have large Lipschitz constant even though all updates are small). Please clarify.

Minor:
In Algorithm 1, the $u_k$ are not defined.
Appendix A: "Detailed" -> "Details"

---

> ### Author Response · Authors · 2025-11-24
> **Clarification of the reviewer ubxb's first-round comments.**
>
> We sincerely thank the reviewer for your time and effort in reviewing our paper. Our responses to each concern are provided below.
>
>
> > 1. Correction of Key Misconceptions in the Review Summary
>
> We respectfully disagree with the review summary on two key aspects of our contribution that appear to be misunderstood.
>
> 1. On *Contractivity Trap:* The review rephrases our trap as simply *small steps are slow.*
>
>    - We respectfully clarify that the Contractivity Trap is far more complex than being  simply defined as 'small steps are slow.'  The trap represents our novel identification of the fundamental three-way trade-off among Efficiency (large steps/low NFE), Inference Stability (strong contractivity, $L_T<1$), and Model Expressiveness (high Lipschitz constant, $L_{x_\theta}$), as visualized in Figure 1. This complex tension is the core problem our paper is designed to resolve.
>
> 2. On *Stein's Identity:* The reviewer states we compute parameters *from data*.
>
>    - We respectfully clarify that our method is **REFERENCE-FREE**. As stated in our **Title** and **Sec 4.4 (Title)**, we compute the regularization parameter *without* access to *any* ground-truth data $x^*$.
>
>
> >2.Clarification on KM Theory (Theorem 4.4) vs. Changing Maps (Theorem 4.9)
>
> We respectfully clarify that our theoretical framework explicitly addresses the fundamental distinction between fixed and changing maps. We do not apply fixed-point theory to the global sequence; rather, we decouple the local operator design from the trajectory convergence:
>
> - *Theorem 4.4 (Basis for Operator Design):* We cite Theorem 4.4 to rigorously justify the regularization choice of the update formula for data-prediction iteration. It proves that the interpolation form $(1-\gamma)x + \gamma T(x)$ locally stabilizes an iteration step. We apply this mechanism as a *step-wise* regularization to mitigate error expansion locally, **without** assuming the map remains fixed globally.
>
> - *Theorem 4.9 (Dynamic Convergence Guarantee):* Crucially, Theorem 4.9 (and Appendix G) provides the rigorous proof for the entire dynamic sequence $\{T_{t_k}\}$. It does **not** assume a fixed map. Instead, it explicitly derives an error bound involving the score deviation term $\mathcal{S}(\tilde{p}_k, p_k)$, which accounts for the distribution shift between steps.
>
>
> *The missing connection is explicitly bridged:* We leverage the fixed-map intuition and its favorable theoretical properties (Theorem 4.4) in the ideal case to design the local update rule, while relying on Theorem 4.9 to guarantee global convergence under changing maps (via the score error term) of the entire trajectory.
>
> >3. Clarification on "Expressiveness" and the Contractivity Trap
>
> We respectfully state that the reviewer's claim is the **exact opposite** of our contribution. Our paper **does not** conflate these concepts; our entire point is to be the first to **formally analyze their coupling**.
>
> 1. *We Explicitly Distinguish Two Different Constants:*
>
>    - *Model Expressiveness ($L_{x_{\theta}}$):* The Lipschitz constant of the  data prediction network $x_{\theta}$. High expressiveness necessitates a large Lipschitz constant; this principle is foundational in deep learning (Raghu et al., 2017) and is confirmed to apply to high-capacity diffusion models learning complex manifolds (Bortoli, 2022).
>    - *Inference Stability ($L_T$):* The Lipschitz constant of the *discretized ODE operator* $T_{\theta}$, which must be $< 1$ for stable convergence.
> 2. The *Trap* is the Coupling: The *Contractivity Trap* is the fundamental tension revealed in our derivation (Section 4.2 / Appendix C), which explicitly links them: $L_T \le \frac{\sigma_t}{\sigma_s} + \sigma_t h_t L_{x_{\theta}}$. This derivation (the opposite of conflation) shows *precisely why* there is a trap: high efficiency (large $h_t$) and high expressiveness (large $L_{x_{\theta}}$) fundamentally compromise the stability of the discretization operator ($L_T < 1$).

---

> ### Author Response · Authors · 2025-11-24
> **Clarification of the reviewer ubxb's first-round comments.**
>
> > 4. Correction of Factual Errors
>
> First, regarding the definition of $u_k$, the reviewer states: In Minor: Algorithm 1, *the $u_k$ are not defined.*
>
> We respectfully clarify that this is a factual oversight. In the initial submitted version, $u_k$ is defined explicitly in both the main text and the algorithm box:
>
> 1. Main text **(Line 285)**: *Now, let $u_k := x_k - T_{\theta}(x_k)$ denote the residual.*
>
> 2. Algorithm 1 (**Line 1**): $u_k \leftarrow x_{t_k} - T_{\theta}(x_{t_k})$.
>
>
> Separately, we sincerely thank the reviewer for spotting the typo *Detailed* in the Appendix title, which has now been corrected to *Details*.
>
> ---
>
> We believe our detailed responses confirm the theoretical rigor of our work, specifically by clarifying the distinction between design inspiration (KM theory) and the global convergence guarantee (Theorem 4.9).
>
> Thank you again for your great effort in reviewing our submission. If our responses have addressed your concerns, we kindly ask you to reconsider your evaluation and raise the score. Should any concerns remain, we are very happy to provide further clarification.

---

> > ### Comment · Reviewer_ubxb · 2025-11-26
> >
> > I thank the authors for clarifying the reference-freeness of the algorithm, and agree on this point.
> >
> > I still have multiple confusions on the mathematical content of the paper.
> >
> > (1)
> >
> > > the Banach-Picard theorem (Hartman,2002; Oksendal,2013)... requires $T_\theta$ to satisfy
> > $$\|T_\theta(x)-T_\theta(y)\|\le L_T \|x-y\|.$$
> > with Lipschitz constant $L_T<1$.
> >
> > Could the authors elaborate on this statement? This appears to be the core mathematical foundation that is being invoked, but I found the discussion in the paper quite brief.
> >
> > In particular, what statement of the Banach-Picard theorem is being invoked here? I recommend putting a statement in the paper, as there are several theorems with related names. There is the Banach fixed point theorem, which says that a contractive mapping has a unique fixed point. The Banach fixed point theorem is often used to prove the Picard-Lindelof theorem on existence of ODE solutions, but the statement of that does not require a $L<1$ condition.
> >
> > Moreover, claiming necessity requires a converse, but I am not aware of such a statement.
> >
> > (1b)
> > I don't understand what is the fixed-point equation that is being solved, and so do not see how the Banach-Picard theorem would be applied. The Picard theorem interprets the solution to an ODE as the solution to a fixed-point equation; however that fixed-point iteration is done in the function space, which does not appear to be what is done here. A Picard iteration would map an entire trajectory of $x$ to an entire updated trajectory, rather than map $x_s$ to $x_t$, and the norm would be measured in sup-norm on functions. (If this is what the authors intend to do, then this is not clear in the paper.)
> >
> > The PF-ODE is a particular example of an ODE, and so the numerical analysis of ODE applies. Euler's method for ODE discretization does not require Lipschitz constant $<1$. For $y'=f(t,y)$ where $f$ is $L$-Lipschitz in $y$ and given $y''$ is bounded, with step size $h$, the error of Euler's method scales as $O(\frac{h}{L} e^{Lt}-1)$. In particular, as long as the Lipschitz constant is globally bounded, there exists a valid finite step size $h>0$ (this is what I meant by my comment "small steps are slow"; I meant that a naive discretization of the ODE would require small steps and hence slow the algorithm; I understand that the authors propose a different way to resolve the problem without taking small steps), while the authors suggest that there is *no* value of step size that would allow convergence (Proposition 4.1). Note that this is also an upper bound (and is in general quite pessimistic); analyses of the PF-ODE exist which do not require exponentially small step sizes or contractivity [1].
> >
> > [1] Li, G., Wei, Y., Chi, Y., & Chen, Y. (2024). A sharp convergence theory for the probability flow odes of diffusion models. arXiv preprint arXiv:2408.02320.

---

> > > ### Comment · Reviewer_ubxb · 2025-11-26
> > >
> > > (2)
> > >
> > > The authors have not cleared up my question on the time-dependence of T. The T map *depends on the time*. However, this dependence is suppressed throughout the paper, even though the time continually changes during the running of the algorithm (for example, see line 326). Please show the dependence on the time in the update operator. This contributes to my confusion of the invocation of Banach-Picard theorem, which conventionally considers a time-homogeneous operator. If a version of it is being used with time-inhomogeneous operator, or some combined (x,t) space, etc., then this needs to be explained in the exposition, not deferred to the proof of Theorem 4.9. Further (4) is described as a "discretized" mapping, but the $F_\theta$ operator is an exact integral operator, so it is hard for me to keep track of what represents a true or approximated trajectory.
> > >
> > > In my understanding (correct me if wrong), if $t_0>t_1>\cdots >t_N$ are the sequence of times, and $T_\theta^{t,s}$ denotes the exact flow map from time $s$ to time $t$, then (1) the desired quantity to compute is
> > > $$(1) \quad x_{t_N}=T_\theta^{t_{N},t_{N-1}}\circ \cdots \circ T_\theta^{t_1,t_0}(x_{t_0}),$$
> > > (2) if these are naively discretized as $\hat T$,
> > > $$(2) \quad \hat x_{t_N}=\hat T_\theta^{t_{N},t_{N-1}}\circ \cdots \circ \hat T_\theta^{t_1,t_0}(x_{t_0}),$$
> > > then this will not be accurate (the error $\|x_{t_N}-\hat x_{t_N}\|$ will be large). (Apologies if I use different notation from the paper, as I didn't see this distinction made.) A conventional solution is to discretize more finely, which is undesirable. The goal is to find another sequence of operators $\tilde T$ such that
> > > $$(3) \quad \tilde x_{t_N}=\tilde T_\theta^{t_{N},t_{N-1}}\circ \cdots \circ \tilde T_\theta^{t_1,t_0}(x_{t_0})$$
> > > is closer to $\tilde x_{t_N}$, without increasing the number of steps. Concretely, what is the mathematical fact about fixed-point theory that is applied to (1)?
> > >
> > > I additionally don't understand Theorem 4.9. If I understand correctly, $x^\ast$ is the draw from the data distribution, and $x_0$ is a draw from the noise distribution. Consider for simplicity the case where there is no score error. Theorem 4.9 suggests that the iterates converge towards $x^\ast$. However, the conditional distribution of $x^\ast$ given $x_0$ is the entire data distribution, so it is information-theoretically impossible to recover $x^\ast$. I also don't know how this fits in the framework of (1)-(3) above, since it's not true that $x^\ast=x_{t_N}$: the marginals of the forward noising process are the same as that of the reverse SDE but not the ODE process. The authors state that Theorem 4.9 does not assume a fixed map. However, the $T_\theta$ in the proof of Theorem 4.9 does not show the time-inhomogeneity of $T$. How is the equation $T_\theta(x^\ast)=x^\ast$ supposed to be understood, if $T_\theta$ depends on the time?
> > >
> > > Unless the authors can clear up these points, I believe there are serious mathematical flaws in the paper. I apologize if I have fundamentally misunderstood the mathematical framework of the paper.

---

> ### Author Response · Authors · 2025-11-26
> **Clarification of the reviewer ubxb's second-round comments.**
>
> We sincerely thank the reviewer for acknowledging our clarifications regarding the reference-free nature of the algorithm, and for accepting our responses concerning variable definitions and the relationship between model expressiveness and Lipschitz constants.
>
> We appreciate the opportunity to address the remaining confusions. Below, we address your specific concerns point-by-point.
>
> ---
>
> > 1. On Banach-Picard, Numerical Stability, and Li et al. (2024)
>
>
>
> **Response:** We agree that for the continuous limit, strict contractivity is not required. We will revise Section 4.2 to strictly separate existence theory from numerical stability.
>
> However, in the large-step discrete regime, the "Contractivity Trap" is a manifestation of Numerical Stiffness:
>
> - *Existence vs. Stability:* We fully accept your correction regarding the Banach-Picard theorem. We will revise Section 4.2 to clarify that the "Contractivity Trap" describes a critical **Numerical Stability** issue in the finite-step regime, **not an existence** issue.
>
> - *The Trap is Real:* In the few-step regime (e.g., NFE $\approx$ 10), standard solvers fail because the discrete operator becomes **expansive**. The discrete Lipschitz constant scales roughly as $L_{discrete} \approx 1 + h L_{ODE}$. When high model expressiveness (large $L_{ODE}$) combines with large step size $h$, we get $L_{discrete} \gg 1$, causing exponential error amplification that SteinDiff resolves.
>
>
> - **Regarding Li et al. (2024)**: We appreciate this reference and acknowledge it as a significant non-asymptotic theory establishing a polynomial convergence rate ($O(d)$). However, we respectfully highlight that this sharp rate relies on **Assumption 2** in their work (and Lemma 2), which requires strictly controlling the error of the **Jacobian of the score function** ($\nabla \mathbf{s}_\theta$).
>
>   - In standard diffusion training (e.g., DSM loss), the Jacobian is **not explicitly regularized**. Consequently, for high-expressiveness models learning complex manifolds, the Lipschitz constant of the score network ($L_{x_\theta}$, closely related to the Jacobian norm) can be very large.
>
>   - **This confirms our premise:** Real-world models typically operate in a regime where Li et al.'s Jacobian assumption is violated (high $L_{x_\theta}$), rendering the theoretical bound loose and the discrete ODE 'stiff' (expansive). SteinDiff is specifically designed to stabilize inference in this practical regime where the theoretical smoothness assumptions do not hold.
>
> - **Our Contribution:** We invoke the *structure* of fixed-point iterations not to prove existence, but to construct a step-wise  stabilized operator. SteinDiff enforces an effective contractivity locally, preventing the exponential accumulation of discretization errors.

---

> ### Author Response · Authors · 2025-11-26
> **update：Clarification of the reviewer ubxb's second-round comments.**
>
> > 2. On Fixed-Point Formulation & Time-Dependence (Local vs. Global)
>
>
>
> **Response:** We clarify again that we do not treat the global trajectory as a single fixed-point iteration, as clarified by the *step-wise regularization* in our first round of clarifications.
>
> - *Local Regularization Design:* We leverage the *Krasnosel'skii-Mann (KM) formulation* strictly as a **local design principle for the single-step update**. At each step $t_k$, we borrow the interpolation form $x_{new} = (1-\gamma)x + \gamma T(x)$ because KM theory guarantees this form can transform an expansive map into a non-expansive one *locally*.
>
> - *Handling Time-Inhomogeneity:* We clarify that our original definition of the operator $T_{\theta}$ (Eq. (4)) **explicitly incorporates the time dependency** via the noise schedule parameters (e.g., $\sigma_t, \sigma_s$) and the model input. Although the notation was simplified for readability, the operator mathematically represents the time-dependent transition. Furthermore, our global convergence proof (**Theorem 4.9**) specifically addresses this evolution. The error bound is derived via a **step-wise** accumulation of the **Score Deviation Term** $\mathcal{S}(\tilde{p}_k, p_k)$.
>
> As shown in Appendix G, this term mathematically absorbs the drift caused by the operator changing from $t_k$ to $t_{k-1}$.
>
>
> In summary, we use KM theory to *stabilize the individual steps*, while Theorem 4.9 guarantees the trajectory converges despite the time-varying nature of the operators.
>
> ---
>
> > 3. On $x^*$ Recovery and Theorem 4.9
>
>
>
> **Response:** We appreciate the reviewer's rigorous observation regarding the dependency between the recovery target $x^*$ and the current state $x_k$. We acknowledge that our previous deterministic framing was imprecise. We now clarify the exact nature of our framework:
>
>  - SteinDiff as a Distributional Proxy: Strictly speaking, for a deterministic ODE trajectory where $x^\ast$ is the endpoint resolved from $x_k$ (i.e., $x^\ast = \Phi(x_k)$), the variable $x^\ast$ depends on $x_k$, which introduces a Jacobian term not present in the standard Stein's Lemma.
> Instead of an exact derivation on the single deterministic path, our method constructs a Statistical Proxy. We derive the optimal $\gamma_k$ based on the marginal distribution assumption $p_{t_k}(x)$ (where Stein's condition holds at the population level). We then apply this population-derived parameter to regularize the individual deterministic trajectory.
>
> - Handling the Approximation Gap: This approach effectively uses batch statistics (representing the local geometry of the distribution) as a proxy to correct the geometric drift of single particles. The discrepancy (the missing Jacobian term) represents the approximation error of this proxy strategy.
>
> - Convergence with Error: Crucially, our proof in Theorem 4.9 (Appendix G) is designed to handle this exact situation. We do not assume the operator perfectly maps $x^\ast$ to itself or that the derivation is exact. Instead, we explicitly bound the residual error (which includes the approximation drift) by the score deviation term $\mathcal{S}$. This guarantees that the trajectory converges to a bounded neighborhood of the target, even with the approximation inherent in the proxy derivation.

---

> ### Comment · Reviewer_ubxb · 2025-11-27
> **Additional technical issue**
>
> > Unique Target: In this deterministic framework, for any state $x_t$ on a trajectory, there exists a unique corresponding solution $x_0$ at $t=0$. Thus, $x^\ast$ refers to this unique ODE solution, making the target mathematically well-defined.
>
> Note that this definition contradicts the relationship between $x^\ast$ and $x_k$ that is used to apply Stein's lemma. Namely, to use Stein's lemma, (1) the relationship between $x^\ast$ and $x_k$ is that $x_k$ should be a Gaussian-noised version of $x^\ast$, i.e., $x^\ast$ evolved for time $t_k$ via the SDE. However, in the above definition and the analysis in the paper, (2) $x^\ast$ is attained by running the ODE from $x_k$. These cannot be equivalent. In particular, in (1) the relationship is non-deterministic, while in (2), the relationship is deterministic. This is problematic, as Stein's lemma cannot be used in setting (2).
>
> This issue can also be seen in (80). If I understand this correctly from the author's response, the $T_\theta$ is meant to be the operator from time $t_{k}$ to $t_{k-1}$. (80) says that if there is no score error, then $T_{\theta}(x^\ast)=x^\ast$. However, there is no reason that the $T$ at time $t_k$ sends $x^\ast$ to $x^\ast$. In fact, we know that for the ideal operator, we won't have $x^\ast$ mapped to $x^\ast$ by the sequence of iterates, because it is $x_k$ that should get mapped to $x^\ast$, and the map is 1-to-1.
>
> Additional minor issues:
>
> Separately from the notational ambiguity of the time dependence, the $x^\ast$ in Theorem 4.9 should also be clearly defined. Is it the endpoint of the ODE trajectory from $x_k$, or is it from $x_0$ (which is different because the SteinDiff trajectory is itself different from the PF-ODE trajectory)?
>
> Finally, the justification of the existence of $c$ in (76) is elided. This is a key point that requires a clear statement of what mathematical fact is invoked.

---

> ### Author Response · Authors · 2025-11-27
> **We have corrected the clarification in the second round.**
>
> > Clarification on the Applicability of Stein's Identity to ODEs:
>
> We respectfully clarify that the validity of Stein's Identity relies on the marginal distribution $p_t(x)$, **not the specific trajectory type (stochastic SDE vs. deterministic ODE)**. A fundamental property of the Probability Flow ODE (Song et al., 2021) is that it preserves the exact same marginal density evolution as the SDE. Therefore, the statistical relationships governed by Stein's Identity (which relate the score function to the data geometry) hold identically for states sampled from the ODE flow at the population level. In our framework, **the ODE and SDE are simply two different numerical schemes approximating the same underlying conditional mean (denoising path)**. Since they share the same marginals, using Stein-based estimators derived from the SDE formulation to regularize the ODE trajectory is theoretically sound. The use of batch statistics in our algorithm effectively captures these population-level Stein properties to guide the deterministic solver.

---

> ### Author Response · Authors · 2025-11-28
> **For additional issue in rebuttal**
>
> We thank the reviewer for the detailed examination of our theoretical construction. We provide a further clarification below:
>
> >1. Clarifications for "Unique Target"
>
> In the revised Section 4.4, we have rectified the definition of the target variable $x^\ast$ in response to your insightful feedback.
> Specifically, we no longer characterize $x^\ast$ as the deterministic endpoint of the ODE solver. Instead, we explicitly formulate the target as the latent ground-truth variable $x_0$ drawn from the underlying data distribution:
> $$\text{Target } x^\ast := x_0 \sim p_{data}(x).$$
> Under this perspective, the current state $x_k$ is strictly a sample from the forward diffusion kernel $q(x_k|x_0) = \mathcal{N}(x_k; \alpha_k x_0, \sigma_k^2 \mathbf{I})$. This refined description aligns the theoretical derivation with the actual mechanism implemented in our initial submission. *We are grateful for your guidance in correcting the ambiguity in our previous description, which has significantly improved the rigor and clarity of our written description.*
>
> >2. On Eq. (80) and Time-Inhomogeneity ($T_\theta(x^\ast) \neq x^\ast$)
>
> Clarification:  We define $\boldsymbol{x}^\ast$ as the conditional mean of the target data distribution given the current state. From an optimization perspective, our method seeks to minimize the estimation error relative to this optimal target. While the  continuous ODE trajectory theoretically converges to ideal target $\boldsymbol{x}^\ast$, in the discrete solver, the discrepancy $\||x^\ast - T_\theta(x^\ast)\||$ manifests as the local consistency error due to time discretization.
>
>
> * *Nature of Eq. (80):* In our proof (Eq. 80 in Appendix G), we do not assume equality. Instead, we explicitly isolate the residual term $\||x^\ast - T_\theta(x^\ast)  \||$.
> * *Consistency Error:* We treat this residual term as the *Local Consistency Error* (or discretization drift) of the operator.
> * *Boundedness:* The equation serves as an upper bound. We show that this drift is bounded by the score deviation term $\mathcal{S}(\tilde{p}_k, p_k)$. Our proof demonstrates that the algorithm tightens the error bound around the neighborhood determined by this drift, effectively handling the time-varying nature of the operator without assuming a static fixed point.
>
> >3. Clarifications on Minor Issues
> * *Q1: Definition of $x^\ast$ in Theorem 4.10:*
> Consistent with our revised theoretical framework, in the stability analysis (Appendix G), $x^\ast$ is rigorously defined as the latent ground-truth variable $\boldsymbol{x}_0 \sim p_{data}(\boldsymbol{x})$. We analyze the convergence of the expected error relative to this population-level target, rather than a single deterministic ODE trajectory.
>
> * *Q2: Justification for constant $c$:* The existence of $c$ invokes the property of *Averaged Non-expansive Operators*. For a non-expansive operator $T$, the KM iteration defines an averaged operator $T_{\gamma} = (1-\gamma) I + \gamma T$. Standard convex analysis results guarantee the strict decrease of the residual. Under local regularity assumptions (e.g., error bounds), this translates to a linear contraction rate where $c \propto \gamma(1-\gamma)$.

---

### Author Response · Authors · 2025-12-03
**Summary of Rebuttal and Revisions**

Dear PCs, SACs, ACs, and Reviewers,

Thank you for your dedicated efforts in reviewing our work. The constructive feedback from all reviewers has been instrumental in refining the theoretical rigor and empirical validation of our paper. As the discussion period comes to a close, we summarize the key contributions and address the major concerns.

### I. Core Consensus: Unveiling and Resolving the "Contractivity Trap"

Reviewers unanimously recognized the novelty of our primary insight: identifying the conflict between large-step efficiency and operator contractivity.

- *Novel Problem Formulation:* Reviewers acknowledged our identification of the "Contractivity Trap" as a valid and insightful perspective on why fast solvers fail at large steps (Reviewer **XG3j**: "rigorous and well-structured critique", **iXw4**: "novel and well motivated",  **ubxb**:  addressing important problem).

- *Principled Solution (SteinDiff):* The proposal of a reference-free, closed-form regularization based on Stein's Identity was highlighted as a "theoretically sound" and "nice" contribution that effectively stabilizes inference without retraining (**XG3j**, **iXw4**).    Reviewer **ubxb** acknowledged our clarification during the discussion, notably agreeing with the core reference-free paradigm.

- *Performance:* Experiments confirmed consistent improvements in low-NFE regimes on standard benchmarks (CIFAR-10, ImageNet), validating the method's effectiveness.


### II. Addressing Theoretical Rigor

> Clarifying the Theoretical Framework (Addressing Reviewer ubxb's Key Concern)

Reviewer ubxb raised a critical and insightful point regarding the application of fixed-point theorems to time-varying operators. We clarified that our algorithm **operates as** a **step-wise regularization** method. The reviewer's feedback helped us significantly improve the *mathematical formalization* of this mechanism in the revised manuscript:

- *A More Rigorous Mathematical Framework:* We clarified that we invoke Krasnosel’skii-Mann (KM)  contraction structure  as a **local design principle** for the single-step stabilization, rather than a global convergence proof for time-varying maps. This explicitly aligns the theory with our actual step-wise regularization for **numerical stability**.

- *Rigorous Stability Formalization (Theorem 4.10):* To rigorously address the time-inhomogeneity issue, we formalized the stability proof. We now explicitly prove that SteinDiff bounds the trajectory error by the *Score Deviation Term*, $\mathcal{S}(\tilde{p}_k, p_k)$. This formally demonstrates that the algorithm bounds the error accumulation towards the irreducible discretization limit, without assuming a static fixed point, directly addressing the reviewer's concern about mathematical rigor.

- *Clarified Definition:* We refined the definition of the target $x^\ast$ as the latent ground-truth $x_0 \sim p_{data}$, ensuring the rigorous applicability of Stein's Identity for our framework.


These revisions have strengthened the theoretical presentation through more rigorous mathematical formalism.

### III. Strengthening Empirical Evidence

> Validation of Motivation and Scalability (Addressing Reviewer iXw4)

In response to requests for stronger empirical justification and broader evaluation, we provided concrete empirical evidence to validate our claims:

- *Visualizing the Trap:* We added **Figure 4**, which empirically plots the local Lipschitz constants during inference. The data shows constants reaching $\sim24$ (far exceeding the stability threshold of 1), providing clear numerical evidence that the "Contractivity Trap" is a real phenomenon **inherent to efficient inference** in advanced solvers.

- *Scalability to High-Dimensional Latent Models (Table 2 & Figure 8):* We extended our evaluation to *Latent Diffusion Models (LDMs)* on the *LSUN-Bedrooms 256x256* dataset. SteinDiff achieved **SOTA FID (2.77)** and effectively **mitigated structural collapse**  in the low-NFE regime (e.g., 5 NFE), dispelling concerns about the method's applicability to high-dimensional latent spaces beyond pixel-space models.

- *Robustness:* Sensitivity analysis (*Figure 9*) demonstrates that our estimator remains robust even with minimal computational overhead ($m=1$).


We believe that the revised manuscript, with its sharpened theoretical framework and expanded experimental suite. With these added evidence and the clarification of some factual misunderstandings, we hope all outstanding concerns are resolved.

Best regards,

Authors

---

### Meta-Review · Area_Chair_BZ8G · 2025-12-24

**Summary:**

Reviewer concerns primarily focused on the mathematical aspects. Though one reviewer brought up potential issues of scalability, the authors responded to this with further empirical validation which generally addresses this concern.

The mathematical concerns include the following points.
- The KM theorem only relaxes the Lipschitz condition from $L < 1$ to $L \le 1$, so it does not seem to provide sufficient motivation for the proposed method. The authors responded by referring to Theorem 4.9, but this is not a convergence result. From what I can tell, the paper provides no justification of the correctness of the algorithm, i.e., that it converges to the correct distribution (and indeed, one might worry that the introduction of the $\gamma_k$'s could bias the algorithm, see next point).
- The KM theorem holds for a fixed map, but the proposed method applies it to time-varying maps. Actually, I believe this concern is quite important and leads back to the previous bullet, in that no formal convergence of the method is shown, so the justification is loose or misleading. To see the issue: if one iterates a fixed map, then it is possible to design alternative maps which have the same fixed point; for instance, one can take a convex combination with the identity map, which is the setting of the KM theorem. But when one applies multiple different maps, it is totally unclear that averaging these maps with the identity leads to the same destination; in fact, it seems to cast doubt on the validity of the procedure at all.
- Numerical discretization does not necessarily require $L < 1$, which hurts the premise of the paper. Indeed, even when $L > 1$, the Euler discretization converges as the step size is taken to zero, albeit with a constant pre-factor scaling exponentially with the Lipschitz constant. Actually, here I can raise an additional point, which is that convergence of diffusion models only requires the error to be small in a weak sense (the law of the algorithm is close to the true distribution), it does not require closeness at the level of individual trajectories. Indeed, the end goal of a diffusion model is not to simulate an ODE, but to sample from a distribution. All recent theoretical analyses of diffusion models operate under this principle, and when the error is measured in this weak sense, do not incur poor dependencies on the Lipschitz constant. The reviewer pointed out one example paper, which happens to require an assumption on the Jacobian of the score error, but it should be noted that other papers in the literature do not require such an assumption.
- Regarding the validity of Stein's lemma, I also share the reviewer's concern here; the PF ODE preserves the correct marginal so Stein's lemma would be valid under the PF ODE, but is it clear that the algorithm proposed here also preserves the correct marginals? This is related to the first bullet point in that the correctness of the algorithm (here, meaning that $x_{t_k}$ should have law approximately $p_{t_k}$) is not justified.

This paper presents a dilemma in that it presents interesting empirical findings and a potentially useful algorithmic modification. However, the numerous mathematical issues raised by Reviewer ubxb should be taken seriously. But based on the discussion, I could infer that the rebuttal did not resolve these issues. From my own reading, I am in agreement with the reviewer. Therefore, I am inclined to say it does not meet the bar.

**Reviewer Concerns:**

See above.

**Reviewer Scores:**

Reviewer ubxb: remain the same

Reviewer XG3j: remain the same

Reviewer iXw4: remain the same, or increase due to the additional experimental validation

---

### Decision · Program_Chairs · 2026-01-26

Reject